# LEARNING LISTWISE DOMAIN-INVARIANT REPRESENTATIONS FOR RANKING

## ABSTRACT

Domain adaptation aims to transfer the knowledge acquired by models trained on (data-rich) source domains to (low-resource) target domains, for which a popular method is invariant representation learning. While they have been studied extensively for problems including classification and regression, how they would apply to ranking problems, where the data and metrics follow a list structure, is not well understood. Theoretically, we establish a domain adaptation generalization bound for ranking under listwise metrics such as MRR and NDCG, that naturally suggests an adaptation method via learning listwise invariant feature representations. Empirically, we demonstrate the benefits of listwise invariant representations by experiments for unsupervised domain adaptation on real-world ranking tasks, including passage reranking. The main novelty of our results is that they are tailored to listwise ranking: the invariant representations are learned at the list level rather than at the item level.

## 1 INTRODUCTION

*Learning to rank* applies machine learning to solve ranking problems that are at the core of many everyday products and applications, including and not limited to search engines and recommendation systems (Liu, 2009). The availability of ever-increasing amounts of training data has enabled larger and larger models to improve the state-of-the-art on more ranking tasks. A prominent example is text retrieval and ranking, where neural language models with billions of parameters easily outperform traditional ranking models, e.g., BM25 (Nogueira et al., 2020). But the need for abundant data means that large neural models may not benefit tasks with little to no annotated data, where they could actually fare worse than baselines such as gradient boosted decision trees (Qin et al., 2021).

Techniques for extending the benefits of large models to low-resource domains include *zero-shot learning* and *domain adaptation*. In the former, instead of directly optimizing for the task of interest with limited data, referred to as the target domain, the model is trained on a data-rich source domain that has a similar data distribution. The latter considers the scenario where (abundant unlabeled) data from the target domain is available, which can be leveraged to estimate the domain shift and improve transferability, e.g., by learning invariant feature representations. This setting and its algorithms are studied extensively for problems including classification and regression (Blitzer et al., 2008; Ganin et al., 2016; Zhao et al., 2018). For ranking problems, however, existing methods are mostly limited to specific tasks and applications. In fact, due to the inherent list structure of the metrics and data, theoretical explorations of domain adaptation for ranking are only nascent.

To this end, we provide the first analysis of domain adaptation for listwise ranking[1] via domain-invariant representations (Section 3), building on the foundational work by Ben-David et al. (2007) for domain adaptation in the binary classification setting. One of the results from our theory is that, when the domain shift is small in terms of the Wasserstein distance, a ranking model optimized on the source is transferrable to the target domain, whose performance under metrics such as MRR and NDCG can be bounded.

---

[1] In this paper, by *listwise ranking*, we mean that the data are defined over lists and the ranking metric is listwise. It is not a statement of the ranking model, e.g., whether the predicted rank assignments are obtained in a pointwise, pairwise or listwise manner (Joachims, 2006; Cao et al., 2007), or the interactions between items in the same list are modeled (Pang et al., 2020).

Inspired by our theory, we propose an adversarial training method for learning listwise domain-invariant representations, called ListDA (Section 4), that minimizes the distributional shifts between source and target domains in the feature space for improving generalization on the target domain. Unlike traditional classification and regression settings, in ranking, each input follows a list structure, containing the items to be ranked. The main technical novelty of ListDA is that the invariant representations it learns are of each *list* as a whole rather than the individual *items* they contain. Empirically, we evaluate ListDA for unsupervised domain adaptation on two ranking tasks (Section 5 and Appendix C), including passage reranking—a fundamental task in information retrieval (Craswell et al., 2019)—where the goal is to rerank a list of candidate documents retrieved by a first-stage retrieval model in response to a search query. We adapt T5 neural rerankers (Raffel et al., 2020) fine-tuned on the general domain MS MARCO dataset (Bajaj et al., 2018) to two specialized domains: biomedical and news articles. Our results demonstrate the benefits of invariant representations on the transferability of rankers trained with ListDA.

## 2 PRELIMINARIES

**Learning to Rank.** A ranking problem is defined by a joint distribution over lists[2] $X \in \mathcal{X}$ of items and nonnegative relevance scores $Y = (Y_1, \cdots, Y_\ell) \in \mathbb{R}_{\geq 0}^\ell$. We assume that all lists are length-$\ell$, and the ground-truth scores are a function of the lists, $y(X)$, so that a ranking problem is equivalently defined by a marginal distribution $\mu^X$ of lists along with a scoring function $y : \mathcal{X} \to \mathbb{R}_{\geq 0}^\ell$.

The goal is to train a *ranker* $f : \mathcal{X} \to S_\ell$ that maps each list $x$ to rank assignments $r := f(x) \in S_\ell$, where $r_i$ represents the predicted rank of item $i$, and $S_\ell$ denotes the set of permutations on $\{1, 2, \cdots, \ell\}$, that recover the descending ordering of the relevance scores $y_i$, i.e., $y_i > y_j \iff r_i < r_j$ for all $i \neq j$. The more common setup is to train a scoring function $h : \mathcal{X} \to \mathbb{R}^\ell$ whose output is a list of ranking scores, s.t. $s_i := h(x)_i$ correlates with $y_i$ in each list, and its ordering agrees that of the ground-truth scores. Rank assignments could be obtained from the ranking scores by taking their descending ordering or via probabilistic models (Section 3).

The quality of the predicted ranks is measured by ranking metrics $u : S_\ell \times \mathbb{R}_{\geq 0}^\ell \to \mathbb{R}_{\geq 0}$, which are functions that take as inputs the rank assignments of the list along with the ground-truth relevance scores and output a utility score. Popular *listwise* metrics in information retrieval include reciprocal rank and normalized discounted cumulative gain (Voorhees, 1999; Järvelin & Kekäläinen, 2002):

**Definition 1** (RR). *Suppose the ground-truth relevance scores* $y \in \{0,1\}^\ell$ *are binary, then the reciprocal rank of the rank assignments* $r \in S_\ell$ *is*

$$\mathrm{RR}(r, y) = \max\{r_i^{-1} : 1 \leq i \leq \ell, y_i = 1\} \cup \{0\}.$$

The expectation of RR over the dataset is called mean reciprocal rank (MRR), $\mathbb{E}[\mathrm{RR}(f(X), y(X))]$.

**Definition 2** (NDCG). *The discounted cumulative gain* (DCG) *and the normalized* DCG *(with identity gain functions, w.l.o.g.) of the rank assignments* $r \in S_\ell$ *are*

$$\mathrm{DCG}(r, y) = \sum_{i=1}^\ell \frac{y_i}{\log(r_i + 1)} \quad \text{and} \quad \mathrm{NDCG}(r, y) = \frac{\mathrm{DCG}(r, y)}{\mathrm{IDCG}(y)},$$

*where* $\mathrm{IDCG}(y) = \max_{r' \in S_\ell} \mathrm{DCG}(r', y)$, *the ideal* DCG, *is the maximum* DCG *value of a list and is attained by the descending ordering of the ground-truth* $y_i$'s.

**Domain Adaptation.** The present work studies the adaptation of a scorer from a source domain $(\mu_S^X, y_S)$ to a target domain $(\mu_T^X, y_T)$. When domain shift is small, i.e., $\mu_S^X \approx \mu_T^X$ and $y_S \approx y_T$, scorers trained on the source are expected to be transferrable to the target without the explicit need of labeled target data. Indeed, the target performance is bounded by the source performance in such cases. As an example, for binary classification, we have the following generalization bound for randomized classifiers (Shen et al., 2018):

---

[2] A natural and common choice for the space of lists $\mathcal{X}$ is the $\ell$-times Cartesian product of $\mathbb{R}^k$, $\mathbb{R}^{\ell \times k}$, meaning that each list $x = (x_1, \cdots, x_\ell)$ is a stack of items represented by $k$-dimensional feature vectors. But generally and more abstractly, the lists need not follow this structure; e.g., $\mathcal{X}$ can also be $\mathbb{R}^d$ (not scaling with $\ell$) provided a mechanism to represent lists by fixed-length vectors.

**Theorem 1.** *Let binary classification problems on a source and a target domain be given by joint distributions $\mu_S, \mu_T$ over inputs and labels $(X, Y) \in \mathcal{X} \times \{0, 1\}$. Let $\mathcal{F} \subset [0, 1]^{\mathcal{X}}$ be a class of L-Lipschitz predictors, and define the error rate of $f \in \mathcal{F}$ by*

$$\mathcal{E}(f) := \mathbb{E}_{(X,Y)\sim\mu}[\mathbb{1}(Y \neq \hat{Y})] := \mathbb{E}_{(X,Y)\sim\mu}[f(X) \cdot \mathbb{1}(Y \neq 1) + (1 - f(X)) \cdot \mathbb{1}(Y \neq 0)],$$

*meaning that the output classifications are probabilistic according to $\mathbb{P}(\hat{Y} = 1 \mid X = x) = f(x)$. Define $\lambda^* := \min_{f'}(\mathcal{E}_S(f') + \mathcal{E}_T(f'))$, then for all $f \in \mathcal{F}$,*

$$\mathcal{E}_T(f) \leq \mathcal{E}_S(f) + 2L \cdot W_1(\mu_S^X, \mu_T^X) + \lambda^*,$$

*where $\mu^X$ denotes the marginal distribution of the input $X$.*

The domain shift in Theorem 1 is measured by the Wasserstein-1 distance between source and target marginal input distributions, whose Kantorovich-Rubinstein dual form is given by (Edwards, 2011):

**Definition 3** (Wasserstein-1). *Let $p, q$ be probability measures on a metric space $(X, d_{\mathcal{X}})$, their Wasserstein-1 distance is $W_1(p, q) = \sup_{f \in \text{Lip}(1)}(\int_{\mathcal{X}} f(x) \, \mathrm{d}p(x) - \int_{\mathcal{X}} f(x) \, \mathrm{d}q(x))$.*

Where the supremum is taken over 1-Lipschitz functionals $f : \mathcal{X} \to \mathbb{R}$:

**Definition 4** (Lipschitz). *Let $(\mathcal{X}, d_{\mathcal{X}}), (\mathcal{X}', d_{\mathcal{X}'})$ be metric spaces. A function $f : \mathcal{X} \to \mathcal{X}'$ is L-Lipschitz, denoted by $f \in \text{Lip}(L)$, if $d_{\mathcal{X}'}(f(x_1), f(x_2)) \leq Ld_{\mathcal{X}}(x_1, x_2)$ for all $x_1, x_2 \in \mathcal{X}$.*

## 3 DOMAIN ADAPTATION GENERALIZATION BOUND FOR RANKING

We establish the first domain adaptation generalization bound for ranking problems under *listwise* ranking metrics. Specifically, we consider the setting of learning scoring functions and (transferrable) representations, where the end-to-end scorer $f = h \circ g$ is a composition of a shared feature map $g : \mathcal{X} \to \mathcal{Z}$ and a scoring function $h : \mathcal{Z} \to \mathbb{R}^\ell$ on the learned list representations. For instance, if the end-to-end scorer is an $m$-layer MLP, we could treat the first $(m - 1)$ layers as $g$ and the last as $h$.

For the bound, we let the rank assignments $r \in S_\ell$ be generated from the output scores $s := h \circ g(x)$ probabilistically via a *Plackett-Luce model* (Plackett, 1975; Luce, 1959), with the exponentiated scores $\exp(s_i)$ as its parameters (Cao et al., 2007; Guiver & Snelson, 2009).

**Definition 5** (P-L model). *A Plackett-Luce model with parameters $v \in \mathbb{R}^\ell_{>0}$ specifies a distribution over $S_\ell$, and the probability mass function, denoted by $p_v$, is defined for all $r \in S_\ell$ by*

$$p_v(r) = \prod_{i=1}^{\ell} \frac{v_{I(r)_i}}{\sum_{j=i}^{\ell} v_{I(r)_j}},$$

*where $I(r)_i$ is the index of the item ranked at $i$, so that $r_{I(r)_i} = i, \forall i$.*

With the above probabilistic procedure of generating rank assignments, the performance (or utility) of a scorer $f$ w.r.t. a ranking metric $u : S_\ell \times \mathbb{R}^\ell_{\geq 0} \to \mathbb{R}_{\geq 0}$ is evaluated by

$$\mathcal{E}(f) := \mathbb{E}_{X\sim\mu^X}\left[\max_{r \in S_\ell} u(r, y(X)) - \mathbb{E}_{R\sim p_{\exp(f(X))}}[u(R, y(X))]\right],$$

which computes its suboptimality relative to the maximum attainable utility. The randomization of the rank assignments using the P-L model is analogous to that of the classifications (using the Bernoulli model) in Theorem 1, and its purpose, together with the exponentiation, is to make the ranking metrics continuous w.r.t. to the raw scores output by the model.

Our analysis, however, differs from that of Theorem 1 due to difficulties arising from the list structure of listwise ranking metrics. For instance, in the realizable setting, the tightness of Theorem 1 hinges on the uniqueness of the optimal classifier. The optimal ranker on ranking problems, in contrast, is generally nonunique. Consider a list with scores $y = (1, 1, 0)$ as an example: the maximum utility is attained by both rank assignments $r = (1, 2, 3)$ and $(2, 1, 3)$. To proceed, we require the following Lipschitz assumptions.

**Assumption 1.** *The ranking metric $u : S_\ell \times \mathbb{R}_{\geq 0}^\ell \to \mathbb{R}_{\geq 0}$ is bounded by $B$ and (Euclidean) $L_u$-Lipschitz in the second argument—the ground-truth relevance scores $y$.*

**Assumption 2.** *The ground-truth scoring function $y : \mathcal{X} \to \mathbb{R}_{\geq 0}^\ell$ is $L_y$-Lipschitz (Euclidean on the output space).*

We will show that RR and NDCG satisfy Assumption 1. Assumption 2 says that similar lists (i.e., close in $\mathcal{X}$) should have similar ground-truth scores. It is satisfied, for instance, when $\mathcal{X}$ is finite and the scores are bounded (this argument is used in Corollary 3); this setup is typical with text data, where the inputs are sequences of one-hot vocabulary encodings.

**Assumption 3.** *The space of input lists $\mathcal{X}$ is a metric space, and the class $\mathcal{H}$ of scoring functions $h : \mathcal{Z} \to \mathbb{R}^\ell$ is $L_h$-Lipschitz (Euclidean on the output space).*

**Assumption 4.** *The feature space $\mathcal{Z}$ is a metric space, and the class $\mathcal{G}$ of feature maps $g : \mathcal{X} \to \mathcal{Z}$ is such that, $\forall g \in \mathcal{G}$, the restrictions of $g$ to the supports of $\mu_S^X$ and $\mu_T^X$, $g|_{\mathrm{supp}(\mu_S^X)}$ and $g|_{\mathrm{supp}(\mu_T^X)}$ respectively, are both invertible with $L_g$-Lipschitz inverses.*

Assumption 3 is standard in generalization and complexity analyses and could be enforced with e.g. $L^2$-regularization (Anthony & Bartlett, 1999; Bartlett et al., 2017). The last assumption is technical, which says that the original inputs are recoverable from their feature representations via Lipschitz $g^{-1}$, meaning that the feature map $g$ should retain as much information from the inputs on each domain. Note that this assumption does not hinder domain-invariant representation learning; as long as $\mathcal{G}$ is sufficiently expressive, $\exists g \in \mathcal{G}$ satisfying $\mu_S^Z = \mu_T^Z$.

We are now ready to state our domain adaptation generalization bound for learning to rank:

**Theorem 2.** *Under Assumptions 1 to 4, for any $g \in \mathcal{G}$, define $\lambda_g^* := \min_{h'}(\mathcal{E}_S(h' \circ g) + \mathcal{E}_T(h' \circ g))$, then for all $h \in \mathcal{H}$,*

$$\mathcal{E}_T(h \circ g) \leq \mathcal{E}_S(h \circ g) + 2(2L_u L_y L_g + B L_h \sqrt{\ell}) \cdot W_1(\mu_S^Z, \mu_T^Z) + \lambda_g^*,$$

*where $\mu^Z$ denotes the marginal distribution of the feature $Z$, $\mu^Z(z) := \mu^X(g^{-1}(z))$.*

The bound says that if the features computed by $g$ are domain-invariant, i.e., $\mu_S^Z = \mu_T^Z$, and the optimal joint risk $\lambda_g^*$ remains low, then a scorer $h$ optimized for the source domain will also perform well on the target. This suggests that domain adaptation for ranking can be achieved via listwise invariant representation learning, where we optimize the feature map $g$ using (unlabeled) source and target domain data to minimize $W_1(\mu_S^Z, \mu_T^Z)$ by aligning the list feature distributions, and simultaneously optimize $h$ on the source domain with labeled data. We propose such a method in Section 4. Indeed, generalization bounds of the type of Theorems 1 and 2 (originally derived by Ben-David et al. (2007)) form the basis of a family of domain adaptation methods based on invariant representation learning, which has seen empirical success in fields ranging from vision (Zhao et al., 2022) to language (Ramponi & Plank, 2020).

The key distinction of our bound is that the features $Z$ are defined on each list $X$ as a whole, rather than the individual items contained in the lists. To illustrate, suppose the setup where the feature representation of each list is a stack of $\ell$ $k$-dimensional vectors, $g(x) = z = (u_1, \cdots, u_\ell) \in \mathbb{R}^{\ell \times k}$, and $u_i \in \mathbb{R}^k$ corresponds to the $i$-th item in the list. For invariant representation learning, we aim for the distributional alignment of not just the items but also the lists, $\mu_S^Z = \mu_T^Z$. Concretely, denote the distribution over item feature vectors by $\mu^U(u) := \mathbb{P}_{Z \sim \mu^Z}(u \in Z)$, then $\mu_S^Z = \mu_T^Z \implies \mu_S^U = \mu_T^U$, but the converse is not true! We demonstrate this point empirically in Section 5, that listwise invariant representations are more appropriate for domain adaptation for listwise ranking, where the data and metrics all follow a list structure, compared to the pointwise method of learning invariant representations of items.

Lastly, to instantiate our bound on MRR and NDCG, we simply verify their Lipschitzness:

**Corollary 3** (Bound for MRR)**.** RR *is $1$-Lipschitz in $y$, thereby*

$$\mathbb{E}_T[\mathrm{RR}(h \circ g)] \geq \mathbb{E}_S[\mathrm{RR}(h \circ g)] - 2(2L_y L_g + L_h \sqrt{\ell}) \cdot W_1(\mu_S^Z, \mu_T^Z) - \lambda_g^*,$$

*where for brevity we wrote $\mathbb{E}[\mathrm{RR}(h \circ g)] := \mathbb{E}_{X \sim \mu^X, R \sim p_{\exp(h \circ g(X))}}[\mathrm{RR}(R, y(X))]$.*

**Corollary 4** (Bound for NDCG). *Suppose $U_{\min} \leq \mathrm{IDCG}(y) \leq U_{\max}$ for some $U_{\min}, U_{\max} \in (0, \infty)$ and all $y \in y_S(\mathrm{supp}(\mu_S^X)) \cup y_T(\mathrm{supp}(\mu_T^X))$, then NDCG is $\widetilde{O}(\sqrt{\ell})$-Lipschitz in $y$, thereby*

$$\mathbb{E}_T[\mathrm{NDCG}(h \circ g)] \geq \mathbb{E}_S[\mathrm{NDCG}(h \circ g)] - \widetilde{O}(\sqrt{\ell}(L_y L_g + L_h)) \cdot W_1(\mu_S^Z, \mu_T^Z) - \lambda_g^*.$$

*where for brevity we wrote $\mathbb{E}[\mathrm{NDCG}(h \circ g)] := \mathbb{E}_{X \sim \mu^X, R \sim p_{\exp(h \circ g(X))}}[\mathrm{NDCG}(R, y(X))].$*

## 4 LEARNING LISTWISE DOMAIN-INVARIANT REPRESENTATIONS

As discussed in Section 3, Theorem 2 suggests that domain adaptation for ranking could be achieved with listwise invariant representations via learning a feature map $g \in \mathcal{G}$ that minimizes the distributional shifts between source and target on the feature space $\mathcal{Z}$, as measured by $W_1(\mu_S^Z, \mu_T^Z)$. Specifically, we consider the setup where the feature representation $z := g(x)$ of each input list containing $\ell$ items, $x = (x_1, \cdots, x_\ell)$, is the stacking of $\ell$ feature vectors, i.e., $\mathcal{Z} = \mathbb{R}^{\ell \times k}$, and each $z := (u_1, \cdots, u_\ell)$ where $u_i \in \mathbb{R}^k$ is the learned feature vector of the $i$-th item in the list. This setup is standard in many learning to rank implementations, e.g., in neural text ranking, each feature vector is an embedding of the input text computed by a language model (Guo et al., 2020).

Learning invariant representations is similar to generative modeling, and a well-known technique in GAN literature is adversarial training (Goodfellow et al., 2014; Ganin et al., 2016), which solves a minimax problem $\max_g \min_{f_{\mathrm{ad}}} \mathcal{L}_{\mathrm{ad}}(g, f_{\mathrm{ad}})$ of two players—the feature map $g$ and an adversary $f_{\mathrm{ad}} : \mathcal{Z} \to \mathbb{R}$. The objective is defined with an adversarial loss function $\ell_{\mathrm{ad}} : \mathbb{R} \times \{0, 1\} \to \mathbb{R}$, whose inputs are the adversary output $\hat{a} := f_{\mathrm{ad}}(z)$ and the domain identity $a$ (set to 1 for target):

$$\mathcal{L}_{\mathrm{ad}}(g, f_{\mathrm{ad}}) := \mathbb{E}_{x \sim \mu_S^X}[\ell_{\mathrm{ad}}(f_{\mathrm{ad}} \circ g(x), 0)] + \mathbb{E}_{x \sim \mu_T^X}[\ell_{\mathrm{ad}}(f_{\mathrm{ad}} \circ g(x), 1)].$$

The adversarial loss corresponds to probability metrics between $\mu_S^Z, \mu_T^Z$ under specific choices of $\ell_{\mathrm{ad}}$ (Goodfellow et al., 2014; Arjovsky et al., 2017). With the 0-1 loss of $\ell_{\mathrm{ad}}(\hat{a}, a) = (1 - a) \cdot \mathbb{1}(\hat{a} \geq 0) + a \cdot \mathbb{1}(\hat{a} < 0)$, $\mathcal{L}_{\mathrm{ad}}$ becomes the (balanced total) classification error of $f_{\mathrm{ad}}$ as a *domain discriminator* on predicting the domain identities, $\mathcal{L}_{\mathrm{ad}}(g, f_{\mathrm{ad}}) = \mathbb{P}_{x \sim \mu_S^X}(f_{\mathrm{ad}} \circ g(x) \geq 0) + \mathbb{P}_{x \sim \mu_T^X}(f_{\mathrm{ad}} \circ g(x) < 0)$, and it gives an upper bound on $W_1(\mu_S^Z, \mu_T^Z)$ under optimality of $f_{\mathrm{ad}}$:

**Proposition 5.** *Denote the metric on $\mathbb{R}^{\ell \times k}$ by $d$, and define $B := \sup_{(z, z') \in \mathrm{supp}(\mu_S^Z \times \mu_T^Z)} d(z, z')$. If $\ell_{\mathrm{ad}}$ is the 0-1 loss, then $W_1(\mu_S^Z, \mu_T^Z) \leq B(1 - \min_{f_{\mathrm{ad}}} \mathcal{L}_{\mathrm{ad}}(g, f_{\mathrm{ad}})).$*

In practice, the 0-1 loss is replaced by a surrogate loss when training $f_{\mathrm{ad}}$ to minimize the classification error, for which we use the logistic loss in our experiments:

$$\ell_{\mathrm{ad}}(\hat{a}, a) = \log(1 + e^{(1-2a)\hat{a}}). \tag{1}$$

Finally, by parameterizing the function class $\mathcal{F}_{\mathrm{ad}} \ni f_{\mathrm{ad}}$, e.g., neural networks, the minimax problem can be optimized with gradient descent-ascent (w.r.t. $g$ and $f$ respectively), implemented with a gradient reversal layer on top of $g$ (Ganin et al., 2016). To prevent $g$ from converging to trivial solutions like $x \mapsto 0$ and causing information loss in the learned features (thereby increasing the minimum achievable $\mathcal{E}_S$ and $\lambda_g^*$), $g$ is optimized together with the scorer $h$ under a joint objective:

$$\mathcal{L}_{\mathrm{joint}}(h, g) = \min_{h \in \mathcal{H}, g \in \mathcal{G}} \left( \mathcal{L}_{\mathrm{rank}}(h \circ g) - \lambda \min_{f_{\mathrm{ad}} \in \mathcal{F}_{\mathrm{ad}}} \mathcal{L}_{\mathrm{ad}}(g, f_{\mathrm{ad}}) \right),$$

where $\mathcal{L}_{\mathrm{rank}}$ is a ranking loss of choice (surrogate to the ranking metric), and $\lambda > 0$ is a hyperparameter controlling the strength of domain-invariant feature learning.

**Choosing $\mathcal{F}_{\mathrm{ad}}$.** The only missing piece is the choice of the discriminator function class. Unlike prior work, however, the distributions $\mu_S^Z, \mu_T^Z$ being modeled here are defined not over items but rather lists (or sets, to be more precise), $z = (u_1, \cdots, u_\ell)$, hence common choices e.g. MLP may not be appropriate. Indeed, a possible design choice is to flatten each list into a single $\ell k$-dimensional vector and feed into an MLP discriminator, but it does not capture the permutation-invariance property of the lists $z$; a list and its permutations are perceived as distinct inputs by the MLP discriminator under this design, despite them being identical as far as the ranker $h$ is concerned. Without permutation-invariance built-in, the optimization of $f_{\mathrm{ad}}$ is data inefficient.

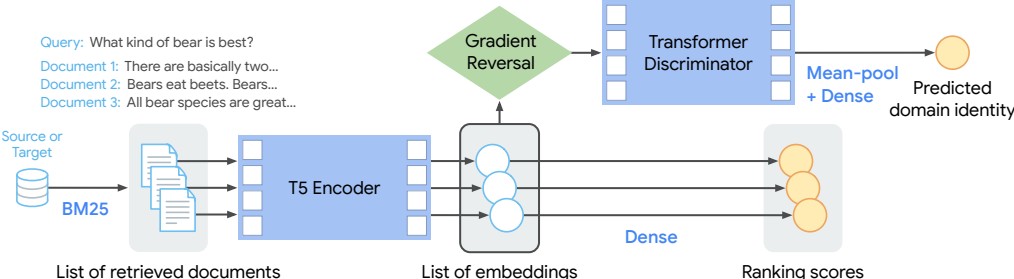

Figure 1: Block diagram of ListDA instantiated on the cross-attention T5 ranker for text ranking.

Therefore, tailored to list-like inputs, we use transformers (no positional encoding) with mean-pooling for $\mathcal{F}_{\text{ad}}$ as a novelty in our implementation (Vaswani et al., 2017), which is permutation-invariant, continuously differentiable, and has good expressive power. Fig. 1 includes a block diagram of our method instantiated on the RankT5 model in Section 5, referred to as ListDA.

## 5 EXPERIMENTS ON PASSAGE RERANKING

In this section, we evaluate ListDA on the passage[3] reranking task, where given a text query, the goal is to rank candidate passages in a retrieved set based on their relevance to the query. In Appendix C, we additionally evaluate ListDA on the ranking task from the Yahoo! Learning to Rank Challenge. Reranking is employed in scenarios where the corpus is too large for all (millions of) documents to be exhaustively ranked by more accurate but expensive models such as SOTA cross-attention rankers based on language models; rather, a simpler but efficient first-stage model such as sparse or dense retrievers e.g. BM25 and DPR is used to retrieve a candidate set of (hundreds or thousands of) passages (Robertson & Zaragoza, 2009; Karpukhin et al., 2020), whose ranks are then refined and improved by a more sophisticated reranking model.

We consider unsupervised domain adaptation. Training neural rerankers requires large amounts of queries and document-relevance annotations. While such data can be obtained relatively easily from search engines under weak supervision for general text domains, annotations on specialized domains such as scientific literature are costly to acquire. Since unannotated documents are almost always readily available, this makes for a suitable candidate to apply unsupervised domain adaptation: specialized rerankers are adapted from ones trained on general domains.

**Models.** We use BM25 as the first-stage retriever for simplicity and focus on the adaptation of the reranker—a cross-attention model based on T5 Base with 250 million parameters (Zhuang et al., 2022). Given a query $q$ and documents $d_1, \cdots, d_\ell$ retrieved by BM25, the list is formed by concatenating the query and each document (with the title, if available), $x = ([q, d_1], [q, d_2], \cdots, [q, d_\ell])$. We follow the setup in Section 4 and use T5 encoder as the feature map $g$, so that the feature representation of the list is the first-token output embeddings that T5 computes on each query-document (q-d) pair,[4] $z = (u_1, \cdots, u_\ell) = (\text{T5}(x_1), \cdots, \text{T5}(x_\ell)) \in \mathbb{R}^{\ell \times 1024}$. The ranking scores are then obtained by projecting each q-d embedding by a dense layer, $s = (h(u_1), \cdots, h(u_\ell))$. We train the ranking model by minimizing the listwise softmax cross-entropy ranking loss:

$$\ell_{\text{rank}}(s, y) = -\sum_{i=1}^{\ell} y_i \log\left(\frac{\exp(s_i)}{\sum_{j=1}^{\ell} \exp(s_j)}\right).$$

ListDA adversarial training also follows Section 4. The discriminator $f_{\text{ad}}$ is a stack of three transformer blocks with the same architecture as those of the T5 encoder. Given a list feature $z = (u_1, \cdots, u_\ell)$, we obtain its domain prediction by feeding all vectors $u_i$ through the transformer blocks at once as a sequence, taking the mean-pool of the outputs and projecting it to a logit with a

---

[3]The terms *document, text,* and *passage* are used interchangeably here.

[4]Under our setup, the feature vectors are computed on each item independently, but generally and ideally $g$ would also model the interactions between items in the same list (Pang et al., 2020).

Table 1: Transfer performance of T5 reranker on top 1000 BM25-retrieved passages.

| Target Domain | Method | MAP | MRR@10 | NDCG@5 | NDCG@10 | NDCG@20 |
|---|---|---|---|---|---|---|
| Robust04 | BM25 | 0.2282 | 0.6801 | 0.4396 | 0.4088 | 0.3781 |
| | Zero-shot | 0.2759 | 0.7977[†] | 0.5857[†] | 0.5340[†] | 0.4856[†] |
| | QGen PL | 0.2693 | 0.7644 | 0.5406 | 0.5034 | 0.4694 |
| | ItemDA | 0.2822[*†] | 0.8037[†] | 0.5822[†] | 0.5396[†] | 0.4922[†] |
| | ListDA | **0.2901**[*†‡] | **0.8234**[*†] | **0.5979**[*†‡] | **0.5573**[*†‡] | **0.5126**[*†‡] |
| TREC-COVID | BM25 | 0.2485 | 0.8396 | 0.7163 | 0.6559 | 0.6236 |
| | Zero-shot | 0.3083 | 0.9217 | 0.8328 | 0.8200 | 0.7826 |
| | QGen PL | 0.3180[*‡] | 0.8907 | 0.8373 | 0.8118 | 0.7861 |
| | ItemDA | 0.3087 | 0.9080 | 0.8276 | 0.8142 | 0.7697 |
| | ListDA | **0.3187**[*‡] | **0.9335** | **0.8693**[*‡] | **0.8412**[†‡] | **0.7985**[‡] |
| BioASQ | BM25 | 0.4088 | 0.5612 | 0.4580 | 0.4653 | 0.4857 |
| | Zero-shot | 0.5008[‡] | 0.6465 | 0.5484[‡] | 0.5542[‡] | 0.5796[‡] |
| | QGen PL | 0.5143[*‡] | 0.6551 | 0.5538[‡] | 0.5643[‡] | 0.5915[*‡] |
| | ItemDA | 0.4781 | 0.6383 | 0.5315 | 0.5343 | 0.5604 |
| | ListDA | **0.5191**[*‡] | **0.6666**[*‡] | **0.5639**[*‡] | **0.5714**[*‡] | **0.5985**[*‡] |

Source domain is MS MARCO. Gain function in NDCG is the identity map. [*]Improves upon zero-shot baseline with statistical significance ($p \leq 0.05$) under the two-tailed Student's $t$-test. [†]Improves upon QGen PL. [‡]Improves upon ItemDA.

dense layer. We use an ensemble of five discriminators as in (Elazar & Goldberg, 2018) to reduce the sensitivity to the randomness in the initialization and the training process, $\sum_{i=1}^{5} \mathcal{L}_{ad}(g, f_{ad}^{(i)})$.

**Datasets.** The source domain in our experiments is MS MARCO for passage ranking, a large-scale dataset containing 8 million passages from the web that covers a wide range of topics, and 532,761 search query and relevant passage pairs (Bajaj et al., 2018). The target domains are biomedical (TREC-COVID, BioASQ) and news articles (Robust04) (Voorhees et al., 2021; Tsatsaronis et al., 2015; Voorhees, 2005). The data are collected and preprocessed as in the BEIR benchmark (Thakur et al., 2021); their paper also contains statistics of the datasets.

Recall from above that the inputs to our cross-attention model are q-d pairs. However, there are no training queries available on two out of three target domains, so following (Ma et al., 2021), we synthesize training queries on all target domains in a zero-shot manner, via a T5 XL query generator (QGen) trained on MS MARCO relevant q-d pairs. QGen synthesizes each query as a seq-to-seq task given a passage from the target corpus as input, whereby we expect the synthesized queries to be related to the input passages. See Table 7 for samples of QGen q-d pairs.

**Baseline Methods.** We compare ListDA to three baseline methods. In **zero-shot** learning, the reranker is trained on MS MARCO only and directly evaluated on the targets. In **QGen PL**, we treat target domain QGen synthesized q-d as relevant pairs and train the reranker on both MS MARCO and QGen q-d pairs (PL as in these q-d pairs are "pseudolabeled" by QGen). This method underlies several recent works on domain adaptation of text retrievers and rerankers (Ma et al., 2021; Sun et al., 2021; Wang et al., 2022). All adaptation methods, including ListDA, are applied on the target domains separately, i.e., we train a model for each source-target pair.

Lastly, prior work that applies invariant representation learning to domain adaptation all performs feature alignment at the item level (Cohen et al., 2018; Tran et al., 2019; Xin et al., 2022) instead of the list level (our ListDA). Yet, the key message of this paper is that the former is not suitable for listwise ranking. To verify and demonstrate our claim, we also compare ListDA to **ItemDA**, which learns pointwise invariant representations with a three-layer MLP discriminator (no improvements from going larger) whose inputs are the q-d embeddings individually (similar to the DANN model by Ganin et al. (2016)).

Due to space constraints, experiment details including hyperparameter settings and the construction of training example lists are relegated to Appendices B.1 and B.2. We also include case studies and additional results with the pairwise logistic ranking loss and the hybrid method of ListDA + QGen PL in Appendix B.

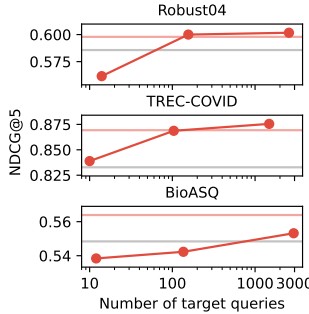 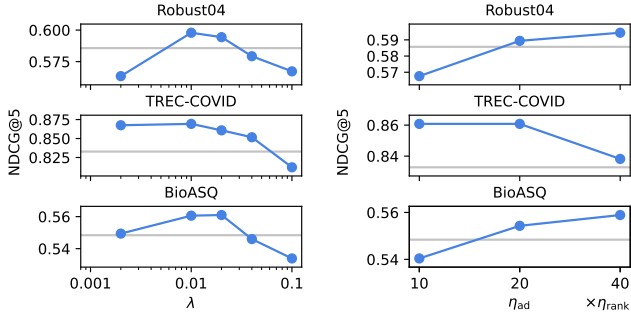

Figure 2: ListDA under different target sizes. Lower grey horizontal line is zero-shot, upper red line is ListDA using all QGen queries.

Figure 3: ListDA performance under different hyperparameter settings for $\lambda$ and $\eta_{ad}$. Grey horizontal line is zero-shot. On the left, $\eta_{ad} = 0.004$ is fixed and $\lambda$ varies. On the right, $\lambda = 0.02$ is fixed and $\eta_{ad}$ varies.

## 5.1 RESULTS

The main results are presented in Table 1. We report metrics are that are commonly used in the literature (e.g., TREC-COVID uses NDCG@20), and evaluate rank assignments given by the descending ordering of the ranking scores. Since TREC-COVID and Robust04 are annotated with 3-level relevancy, the scores are binarized for mean average precision (MAP) and MRR as follows: for TREC-COVID, 0 (not relevant) and 1 (partially relevant) are negative, 2 (fully relevant) is positive; for Robust04, 0 (not relevant) is negative, 1 (relevant) and 2 (highly relevant) are positive.

Across all three datasets, ListDA has the best performance, and the fact that it uses the same resource as QGen PL demonstrates the benefits of invariant representations. Furthermore, the favorable comparison of ListDA to ItemDA supports the discussion in Section 3 that for domain adaptation for listwise ranking, the invariant representations the model learns should be of each list as a whole, and not of the items individually.

**Quality of QGen.** An explanation for why QGen PL underperforms ListDA despite sharing resources is that the negative sampling of irrelevant q-d pairs could lead to false negatives in the training data. Sun et al. (2021) observed that queries synthesized by QGen lack specificity and could be relevant to many documents. While QGen PL trains on the QGen pseudolabels and the randomly sampled irrelevant documents by treating them as ground truth, they are not assumed by ListDA, which is thereby less likely to be affected by false negatives or false positives—when synthesized queries turn out to be irrelevant to the input passages (see Table 9 for samples). While out of the scope, improving query generation should boost the performance of both QGen PL and ListDA.

## 5.2 ANALYSIS OF LISTDA

**Size of Target Data.** Unsupervised domain adaptation requires sufficient unlabeled data, but not all domains have the same amount: BioASQ has 14 million documents (also the total number of QGen queries, as we synthesize one per document), but Robust04 only 528,155, and TREC-COVID 171,332. In Fig. 2, we plot the performance of ListDA under varying numbers of target QGen queries (also the number of target training lists). Surprisingly, on Robust04 and TREC-COVID, using just ~100 target QGen queries (0.03% and 0.06% of all, respectively) is sufficient for ListDA to achieve full performance! Although the number of queries is small, since 1,000 documents are retrieved per query, the total number of distinct target documents is still substantial—up to 100,000, or 29.5% and 60.7% of the respective corpora. Performance begins to drop when reduced to ~10 queries, capping the number of documents at 10,000 (2.7% and 5.8%, respectively). The same data efficiency, however, is not observed on BioASQ, likely due to the hardness of the dataset from e.g. the extensive use of specialized biomedical terms (Tables 7 to 9).

**Sensitivity to Hyperparameters.** ListDA introduces two main hyperparameters for the discriminator $f_{ad}$: the learning rate $\eta_{ad}$ and the strength of invariant feature learning $\lambda$. We plot in Fig. 3 the

sensitivity of their settings by fixing one and varying the other. It is observed that a balanced choice of $\lambda$ is needed to elicit the best performance from ListDA, but the same choice largely works well across datasets. We set $\eta_{\mathrm{ad}}$ to be multiples of the reranker learning rate $\eta_{\mathrm{rank}}$, and the results show that each dataset prefers different settings of $\eta_{\mathrm{ad}}$, probably due to their distinct domain characteristics.

## 6 RELATED WORK

**Learning to Rank and Text Ranking.** Traditional learning to rank concerns tabular datasets with numerical features for which a wide array of models are developed (Liu, 2009), ranging from SVMs (Joachims, 2006), gradient boosted decision trees (Burges, 2010), to neural rankers (Burges et al., 2005; Pang et al., 2020; Qin et al., 2021). Another research direction is the design of ranking losses (surrogate to ranking metrics), which are categorized into pointwise, pairwise, and listwise approaches (Cao et al., 2007; Bruch et al., 2020; Zhu & Klabjan, 2020; Jagerman et al., 2022a).

Recent advances in large neural language models have spurred interest in applying them on text ranking tasks (Lin et al., 2022), leading to cross-attention models (Han et al., 2020; Nogueira & Cho, 2020; Nogueira et al., 2020; Pradeep et al., 2021) and generative models based on query likelihood (dos Santos et al., 2020; Zhuang & Zuccon, 2021; Zhuang et al., 2021; Sachan et al., 2022). A different line of work is neural text retrieval models, which emphasizes efficiency, including dual-encoder (Karpukhin et al., 2020; Zhan et al., 2021), late-interaction (Khattab & Zaharia, 2020; Hui et al., 2022), and models based on transformer memory (Tay et al., 2022).

**Domain Adaptation.** Following (Ben-David et al., 2007; Blitzer et al., 2008), a family of domain adaptation methods is based on learning (adversarial) domain-invariant feature representations (Long et al., 2015; Ganin et al., 2016; Courty et al., 2017). These methods are applied in fields including NLP and on tasks ranging from cross-domain sentiment analysis, question-answering (Li et al., 2017; Vernikos et al., 2020), to unsupervised cross-lingual learning and machine translation (Xian et al., 2022; Lample et al., 2018). Our method also belongs to this family, but to the best of our knowledge, no prior work considers learning domain-invariant representations of lists/sets.

Recently, Zhao et al. (2019) point out that for classification, Theorem 1 admits a lower bound under perfect feature alignment and source accuracy when there are distribution shifts in class priors between source and target domains. Their result does not apply to ranking problems, and we leave investigations on the effects of shifts in marginal distributions of relevance scores to future work.

**Domain Adaptation for Information Retrieval.** Work on this subject is categorized into supervised and unsupervised domain adaptation. The former assumes access to labeled source data and (a small amount of; few-shot) labeled target data (Sun et al., 2021). The present work focuses on the latter, only assuming access to unannotated target documents. Cohen et al. (2018) apply invariant representation learning to unsupervised domain adaptation for text ranking, followed by Tran et al. (2019) for enterprise email search, and Xin et al. (2022) for dense retrieval. However, their approach learns invariant representations of items and not lists. Another family of approaches is based on query generation (Ma et al., 2021; Wang et al., 2022), initially proposed for dense retrieval.

## 7 CONCLUSION

We analyze domain adaptation for ranking theoretically and establish a generalization bound under listwise ranking metrics. Our bound leads to an adaptation method based on learning listwise domain-invariant representations, called ListDA, which is empirically demonstrated to be effective for unsupervised domain adaptation on various ranking tasks.

The novelty of our results is that they are tailored to listwise ranking, where the data the metrics all follow a list structure. A key message from our theory and experiments is that when applying invariant representation learning to domain adaptation for ranking problems, the representations learned should be at the list level rather than at the item level. We believe our theoretical and empirical contributions provide a foundation for future studies on domain adaptation for ranking.

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

## A  OMITTED PROOFS

Before proving the generalization bounds for binary classification (Theorem 1) and ranking (Theorem 2), recall the following properties of Lipschitz functions:

**Fact 6** (Properties of Lipschitz functions)**.**

1. $f : \mathbb{R}^d \to \mathbb{R}$ is differentiable, then it is (Euclidean) $L$-Lipschitz if and only if $\|\nabla f\|_2 \leq L$.

2. If $f : \mathcal{X} \to \mathbb{R}$ is $L$-Lipschitz and $g : \mathcal{X} \to \mathbb{R}$ is $M$-Lipschitz, then $af + bg$ is $(|a|L + |b|M)$-Lipschitz, and $\max(f, g)$ is $\max(L, M)$-Lipschitz.

3. If $f : \mathcal{X} \to \mathcal{Y}$ is $L$-Lipschitz and $g : \mathcal{Y} \to \mathcal{Z}$ is $M$-Lipschitz, then $g \circ f$ is $LM$-Lipschitz.

*Proof.* For the first statement, suppose bounded gradient norms, then by mean value theorem $\exists t \in [0, 1]$ s.t. $f(y) - f(x) = \nabla f(z)^\top (y - x)$ with $z := (1 - t)x + ty$, so by Cauchy-Schwarz,
$$\|f(y) - f(x)\|_2 \leq \|\nabla f(z)\|_2 \|y - x\|_2 \leq L\|y - x\|_2.$$
Next, suppose $L$-Lipschitzness, then by differentiability, $\nabla f(x)^\top z = f(x + z) - f(x) + o(\|z\|_2)$. Set $z := t\nabla f(x)$, we have
$$t\|\nabla f(x)\|_2^2 = f(x + t\nabla f(x)) - f(x) + o(t\nabla f(x)) \leq Lt\|\nabla f(x)\|_2 + o(t\|\nabla f(x)\|_2),$$
and the result follows by dividing both sides by $t\|\nabla f(x)\|_2$ and taking $t \to 0$.

For the second,
$$|af(x) + bg(x) - (af(y) + bg(y))|$$
$$\leq |a||f(x) - f(y)| + |b||g(x) - g(y)| \leq (|a|L + |b|M)d_\mathcal{X}(x, y).$$
Next, assume w.l.o.g. $\max(f(x), g(x)) - \max(f(y), g(y)) \geq 0$, then
$$|\max(f(x), g(x)) - \max(f(y), g(y))|$$
$$= \begin{cases} f(x) - \max(f(y), g(y)) \leq f(x) - f(y) \leq Ld_\mathcal{X}(x, y) & \text{if } \max(f(x), g(x)) = f(x) \\ g(x) - \max(f(y), g(y)) \leq g(x) - g(y) \leq Md_\mathcal{X}(x, y) & \text{else} \end{cases}$$
$$\leq \max(L, M)d_\mathcal{X}(x, y).$$

For the third, $d_\mathcal{Z}(g \circ f(x), g \circ f(y)) \leq Md_\mathcal{Y}(f(x), f(y)) \leq LMd_\mathcal{X}(x, y)$. $\qquad \square$

We first prove Theorem 1, as it shares the same organization of the arguments with Theorem 2.

*Proof of Theorem 1.* Define random variable $\eta := \mathbb{1}(Y = 1)$, then $\mathcal{E}(f) = \mathbb{E}_{(X,Y)\sim\mu}[\eta - (2\eta - 1)f(X)]$. Note that
$$\mathcal{E}(f) - \mathcal{E}(f') = \mathbb{E}_{(X,Y)\sim\mu}[\eta - (2\eta - 1)f(X)] - \mathbb{E}_{(X,Y)\sim\mu}[\eta - (2\eta - 1)f'(X)]$$
$$= \mathbb{E}_{(X,Y)\sim\mu}[(2\eta - 1) \cdot (f'(X) - f(X))]$$
$$\leq \mathbb{E}_{X\sim\mu^X}[|f'(X) - f(X)|]$$
because $(2\eta - 1) = \pm 1$. On the other hand,
$$\mathbb{E}_{X\sim\mu^X}[|f(X) - f'(X)|] = \mathbb{E}_{(X,Y)\sim\mu}[|(2\eta - 1) \cdot (f(X) - f'(X)) - \eta + \eta|]$$
$$\leq \mathbb{E}_{(X,Y)\sim\mu}[|(2\eta - 1)f(X) - \eta|] + \mathbb{E}_{(X,Y)\sim\mu}[|-(2\eta - 1)f'(X) + \eta|]$$
$$= \mathbb{E}_{(X,Y)\sim\mu}[\eta - (2\eta - 1)f(X)] + \mathbb{E}_{(X,Y)\sim\mu}[\eta - (2\eta - 1)f'(X)]$$
$$= \mathcal{E}(f') + \mathcal{E}(f).$$

Then by Fact 6, the fact that taking absolute value is 1-Lipschitz, and Definition 3, for all $f, f' \in \mathcal{F}$,
$$\mathcal{E}_T(f) = \mathcal{E}_S(f) + (\mathcal{E}_T(f) - \mathcal{E}_T(f')) - (\mathcal{E}_S(f) + \mathcal{E}_S(f')) + (\mathcal{E}_S(f') + \mathcal{E}_T(f'))$$
$$\leq \mathcal{E}_S(f) + \left(\mathbb{E}_{X\sim\mu_T^X}[|f(X) - f'(X)|] - \mathbb{E}_{X\sim\mu_S^X}[|f(X) - f'(X)|]\right) + (\mathcal{E}_S(f') + \mathcal{E}_T(f'))$$
$$\leq \mathcal{E}_S(f) + \sup_{q\in\text{Lip}(2L)} (\mathbb{E}_{X\sim\mu_T^X}[q(X)] - \mathbb{E}_{X\sim\mu_S^X}[q(X)]) + (\mathcal{E}_S(f') + \mathcal{E}_T(f'))$$
$$\leq \mathcal{E}_S(f) + 2L \cdot W_1(\mu_S^X, \mu_T^X) + (\mathcal{E}_S(f') + \mathcal{E}_T(f')).$$
and the result follows by taking the min over $f'$. $\qquad \square$

Next, we prove Theorem 2. The main idea behind the proof is that under our setup and assumptions, we could write $\mathcal{E}_S$ and $\mathcal{E}_T$ as expectations of Lipschitz functions of $Z \sim \mu_S^Z$ and $\mu_T^Z$, respectively, so by Definition 3 their difference can be upper bounded by $W_1(\mu_S^Z, \mu_T^Z)$.

While omitted, Theorem 2 can be extended to the cutoff version of the ranking metric $u$ with a simple modification of the proof. Also, a finite sample generalization bound could be obtained using e.g., Rademacher complexity, assuming Lipschitzness of the end-to-end scorer (Blitzer et al., 2008; Shalev-Shwartz & Ben-David, 2014).

*Proof of Theorem 2.* Fix $g \in \mathcal{G}$, which has a $L_g$-Lipschitz inverse $g^{-1}$ to $\mathrm{supp}(\mu_S^X)$ by Assumption 4. Define $\epsilon_{h,g} : \mathcal{Z} \to \mathbb{R}_{\geq 0}$ for any given $h : \mathcal{Z} \to \mathbb{R}^\ell$ by

$$\epsilon_{h,g}(z) := \max_{r \in S_\ell} u(r, y_S \circ g^{-1}(z)) - \mathbb{E}_{R \sim p_{\exp(h(z))}}[u(R, y_S \circ g^{-1}(z))]$$

$$= \max_{r \in S_\ell} u(r, y_S \circ g^{-1}(z)) - \sum_{r \in S_\ell} u(r, y_S \circ g^{-1}(z)) \prod_{i=1}^{\ell} \frac{\exp(h(z)_{I(r)_i})}{\sum_{j=i}^{\ell} \exp(h(z)_{I(r)_j})},$$

and note that $\mathcal{E}_S(h \circ g) = \mathbb{E}_{X \sim \mu_S^X}[\epsilon_{h,g}(g(X))] =: \mathbb{E}_{Z \sim \mu_S^Z}[\epsilon_{h,g}(z)]$. An analogous analysis holds for $\mathcal{E}_T$.

We show that $\epsilon_{h,g}$ as written above is a Lipschitz function of $z$ if $h$ is Lipschitz. For the first term, because $u$ is $L_u$-Lipschitz in $y_S \circ g^{-1}(z)$ and $y_S \circ g^{-1}(z)$ is $L_y L_g$-Lipschitz in $z$, so $u$ is $L_u L_y L_g$-Lipschitz in $z$, and so is $z \mapsto \max_{r \in S_\ell} u(r, y_S \circ g^{-1}(z))$ by Fact 6. Now we bound the second term. We show that it is Lipschitz in both $y_S \circ g^{-1}(z) =: y$ and $h(z) =: s$ under the Euclidean distance. By Jensen's inequality,

$$\|\nabla_y \mathbb{E}_{R \sim p_{\exp(s)}}[u(R, y)]\|_2 = \|\mathbb{E}_{R \sim p_{\exp(s)}}[\nabla_y u(R, y)]\|_2 \leq \mathbb{E}_{R \sim p_{\exp(s)}}[\|\nabla_y u(R, y)\|_2] \leq L_u.$$

Next,

$$\|\nabla_s \mathbb{E}_{R \sim p_{\exp(s)}}[u(R, y)]\|_2 = \sqrt{\sum_{k=1}^{\ell} \nabla_s \left( \sum_{r \in S_\ell} u(r, y) \prod_{i=1}^{\ell} \frac{\exp(s_{I(r)_i})}{\sum_{j=i}^{\ell} \exp(s_{I(r)_j})} \right)^2_k} \leq B\sqrt{\ell},$$

where the inequality is due to product rule and

$$\left| \frac{\partial}{\partial s_{I(r)_m}} \left( \sum_{r \in S_\ell} u(r, y) \prod_{i=1}^{\ell} \frac{\exp(s_{I(r)_i})}{\sum_{j=i}^{\ell} \exp(s_{I(r)_j})} \right) \right|$$

$$= \left| \sum_{r \in S_\ell} u(r, y) \sum_{i=1}^{\ell} \prod_{k \neq i}^{\ell} \left( \frac{\partial}{\partial s_{I(r)_m}} \frac{\exp(s_{I(r)_i})}{\sum_{j=k}^{\ell} \exp(s_{I(r)_j})} \right) \frac{\exp(s_{I(r)_i})}{\sum_{j=k}^{\ell} \exp(s_{I(r)_j})} \right|$$

$$= \sum_{r \in S_\ell} u(r, y) \sum_{i=1}^{\ell} \mathbb{1}(m \leq i) \left( 1 - \frac{\exp(s_{I(r)_i})}{\sum_{j=k}^{\ell} \exp(s_{I(r)_j})} \right) \prod_{k=1}^{\ell} \frac{\exp(s_{I(r)_i})}{\sum_{j=k}^{\ell} \exp(s_{I(r)_j})}$$

$$\leq B \sum_{r \in S_\ell} \sum_{i=1}^{\ell} \prod_{k=1}^{\ell} \frac{\exp(s_{I(r)_i})}{\sum_{j=k}^{\ell} \exp(s_{I(r)_j})} = B;$$

recall that $\frac{\mathrm{d}}{\mathrm{d}x_i} \mathrm{softmax}(x)_j = \mathrm{softmax}(x)_i(\mathbb{1}(i=j) - \mathrm{softmax}(x)_j)$. Suppose $h \in \mathrm{Lip}(L_h)$, then $z \mapsto \mathbb{E}_{R \sim p_{\exp(h(z))}}[u(R, y_S \circ g^{-1}(z))]$ is Lipschitz because

$$\left| \mathbb{E}_{R \sim p_{\exp(h(z))}}[u(R, y_S \circ g^{-1}(z))] - \mathbb{E}_{R \sim p_{\exp(h(z'))}}[u(R, y_S \circ g^{-1}(z'))] \right|$$

$$\leq \left| \mathbb{E}_{R \sim p_{\exp(h(z))}}[u(R, y_S \circ g^{-1}(z))] - \mathbb{E}_{R \sim p_{\exp(h(z))}}[u(R, y_S \circ g^{-1}(z'))] \right|$$

$$\quad + \left| \mathbb{E}_{R \sim p_{\exp(h(z))}}[u(R, y_S \circ g^{-1}(z'))] - \mathbb{E}_{R \sim p_{\exp(h(z'))}}[u(R, y_S \circ g^{-1}(z'))] \right|$$

$$\leq L_u \|y_S \circ g^{-1}(z) - y_S \circ g^{-1}(z')\|_2 + B\sqrt{\ell}\|h(z) - h(z')\|_2$$

$$\leq (L_u L_y L_g + B L_h \sqrt{\ell}) d_{\mathcal{X}}(x, x').$$

Putting everything together, $\epsilon_{h,g}$ is $(2L_u L_y L_g + BL_h\sqrt{\ell})$-Lipschitz in $z$ for any $h \in \mathrm{Lip}(L_h)$.

Then by Fact 6 and Definition 3, for all $g \in \mathcal{G}$ and $h, h' \in \mathcal{H}$,

$$
\begin{aligned}
\mathcal{E}_T(h \circ g) &= \mathcal{E}_S(h \circ g) + (\mathcal{E}_T(h \circ g) - \mathcal{E}_T(h' \circ g)) - (\mathcal{E}_S(h \circ g) + \mathcal{E}_S(h' \circ g)) \\
&\qquad\qquad\qquad\qquad\qquad\qquad\qquad\qquad + (\mathcal{E}_S(h' \circ g) + \mathcal{E}_T(h' \circ g)) \\
&\leq \mathcal{E}_S(h \circ g) + (\mathcal{E}_T(h \circ g) - \mathcal{E}_T(h' \circ g)) - (\mathcal{E}_S(h \circ g) - \mathcal{E}_S(h' \circ g)) \\
&\qquad\qquad\qquad\qquad\qquad\qquad\qquad\qquad + (\mathcal{E}_S(h' \circ g) + \mathcal{E}_T(h' \circ g)) \\
&= \mathcal{E}_S(h \circ g) + \mathbb{E}_{Z \sim \mu_T^Z}[\epsilon_{h,g}(Z) - \epsilon_{h',g}(Z)] - \mathbb{E}_{Z \sim \mu_S^Z}[\epsilon_{h,g}(Z) - \epsilon_{h',g}(Z)] \\
&\qquad\qquad\qquad\qquad\qquad\qquad\qquad\qquad + (\mathcal{E}_S(h' \circ g) + \mathcal{E}_T(h' \circ g)) \\
&\leq \mathcal{E}_S(h \circ g) + \sup_{q \in \mathrm{Lip}(2(2L_u L_y L_g + BL_h\sqrt{\ell}))} (\mathbb{E}_{Z \sim \mu_T^Z}[q(Z)] - \mathbb{E}_{Z \sim \mu_S^Z}[q(Z)]) \\
&\qquad\qquad\qquad\qquad\qquad\qquad\qquad\qquad + (\mathcal{E}_S(h' \circ g) + \mathcal{E}_T(h' \circ g)) \\
&\leq \mathcal{E}_S(h \circ g) + 2(2L_u L_y L_g + BL_h\sqrt{\ell}) \cdot W_1(\mu_S^Z, \mu_T^Z) + (\mathcal{E}_S(h' \circ g) + \mathcal{E}_T(h' \circ g)),
\end{aligned}
$$

and the result follows by taking the min over $h'$. $\qquad\square$

Finally, we verify the Lipschitz conditions for RR and NDCG.

*Proof of Corollary 3.* It suffices to verify that $y \mapsto \mathrm{RR}(r, y)$ is 1-Lipschitz, which follows from the fact that $\mathrm{RR} \leq 1$ and $\|y - y'\|_2 \geq 1$ for all $y, y' \in \{0,1\}^\ell, y \neq y'$. $\qquad\square$

*Proof of Corollary 4.* It suffices to verify that

$$
y \mapsto \mathrm{NDCG}(r, y) := \frac{\mathrm{DCG}(r,y)}{\mathrm{IDCG}(y)} = \left(\sum_{i=1}^{\ell} \frac{y_i}{\log(r_i^* + 1)}\right)^{-1} \sum_{i=1}^{\ell} \frac{y_i}{\log(r_i + 1)}
$$

is Lipschitz. Note that $\mathrm{IDCG}(y) = \max_r \mathrm{DCG}(r, y)$, a max of continuous functions, is piecewise continuous in $y$ where each piece is defined by an $r' \in S_\ell$: $\{y : r' = \arg\max_r \mathrm{DCG}(r, y)\}$.

Let $r \in S_\ell$, and $y, y' \in \mathbb{R}^\ell$ s.t. $\arg\max_{r'} \mathrm{DCG}(r', y) = \arg\max_{r'} \mathrm{DCG}(r', y') =: r^*$, i.e., they are on the same piece for IDCG. Then

$$
\begin{aligned}
&\left|\frac{\partial}{\partial y_k} \mathrm{NDCG}(r, y)\right| \\
&= \left| \mathrm{IDCG}(y)^{-1} \cdot \frac{\partial}{\partial y_k} \sum_{i=1}^{\ell} \frac{y_i}{\log(r_i + 1)} - \mathrm{DCG}(r,y) \cdot \left(\frac{\partial}{\partial y_k} \sum_{i=1}^{\ell} \frac{y_i}{\log(r_i^* + 1)}\right)^{-2}\right| \\
&\leq \left|\mathrm{IDCG}(y)^{-1} \cdot \log(r_k + 1)^{-1}\right| + \left|\mathrm{DCG}(r,y) \cdot \log(r_k^* + 1)^2\right| \\
&\leq \left|\mathrm{IDCG}(y)^{-1} \cdot \log(2)^{-1}\right| + \left|\mathrm{DCG}(r,y) \cdot \log(\ell + 1)^2\right| \\
&\leq U_{\min}^{-1} + U_{\max}\log(\ell + 1)^2,
\end{aligned}
$$

so NDCG is $\sqrt{\ell}(U_{\min}^{-1} + U_{\max}\log(\ell+1)^2)$-Lipschitz by combining the above and the fact that IDCG is continuous. $\qquad\square$

*Proof of Proposition 5.* First,

$$
\begin{aligned}
W_1(\mu_S^Z, \mu_T^Z) &= \inf_{\gamma \in \Gamma(\mu_S^Z, \mu_T^Z)} \int_{\mathcal{Z} \times \mathcal{Z}} d(z, z') \, \mathrm{d}\gamma(z, z') \leq B \inf_{\gamma \in \Gamma(\mu_S^Z, \mu_T^Z)} \int_{\mathcal{Z} \times \mathcal{Z}} \mathbb{1}(z \neq z') \, \mathrm{d}\gamma(z, z') \\
&= B\left(1 - \sup_{\gamma \in \Gamma(\mu_S^Z, \mu_T^Z)} \int_{\mathcal{Z} \times \mathcal{Z}} \mathbb{1}(z = z') \, \mathrm{d}\gamma(z, z')\right) \\
&= B\left(1 - \int_{\mathcal{Z}} \min(\mu_S^Z(z), \mu_T^Z(z)) \, \mathrm{d}z\right) \\
&= B\int_{\mathcal{Z}} \max(0, \mu_T^Z(z) - \mu_S^Z(z)) \, \mathrm{d}z = \frac{B}{2}\int_{\mathcal{Z}} \left|\mu_T^Z(z) - \mu_S^Z(z)\right| \, \mathrm{d}z,
\end{aligned}
$$

because $\int \mu_T^Z(z) - \mu_S^Z(z) \, \mathrm{d}z = 0$. Note that the last term is the total variation between $\mu_S^Z, \mu_T^Z$.

On the other hand, define $\hat{Y}(z) := \mathbb{1}(f_{\mathrm{ad}}(z) \geq 0)$. Then the balanced total error rate of $\hat{Y}$ on predicting the domain identities is

$$\mathrm{Err}(\hat{Y}) := \int_{\mathcal{Z}} \Big( \hat{Y}(z)\mu_S^Z(z) + (1 - \hat{Y}(z))\mu_T^Z(z) \Big) \, \mathrm{d}z = 1 + \int_{\mathcal{Z}} \Big( \hat{Y}(z) - \frac{1}{2} \Big)\big(\mu_S^Z(z) - \mu_T^Z(z)\big) \, \mathrm{d}z.$$

This quantity is minimized with $\hat{Y}^*(z) = \mathbb{1}(\mu_T^Z(z) \geq \mu_S^Z(z))$, whereby

$$\mathrm{Err}(\hat{Y}^*) = 1 - \frac{1}{2}\int_{\mathcal{Z}} \big|\mu_S^Z(z) - \mu_T^Z(z)\big| \, \mathrm{d}z \leq 1 - \frac{1}{B}W_1(\mu_S^Z, \mu_T^Z).$$

The result follows from an algebraic rearrangement of the terms. $\qquad\square$

## B  ADDITIONAL EXPERIMENTS ON PASSAGE RERANKING AND DETAILS

In this section, additional experiment results for unsupervised domain adaptation on the passage reranking task considered in Section 5 are provided, along with case studies on ListDA vs. zero-shot and QGen PL (Tables 8 and 9), hyperparameter settings (Appendix B.1) and details on the construction of training example lists (Appendix B.2).

**Pairwise Logistic Ranking Loss.**   We experiment with the pairwise logistic ranking loss in place of the listwise softmax cross-entropy loss (Eq. (1)) used in Section 5, defined as

$$\ell_{\mathrm{rank}}(s, y) = -\sum_{i=1}^{\ell}\sum_{j=1}^{\ell} \mathbb{1}(y_i > y_j) \log\Big( \frac{\exp(s_i)}{\exp(s_i) + \exp(s_j)} \Big).$$

The results on the Robust04 dataset are provided in Table 2a. Since the pairwise logistic loss does not perform better than softmax cross-entropy (cf. Table 1; also see (Jagerman et al., 2022b)), further experiments with this ranking loss are not pursued.

As an implementation remark, in this set of experiments, we do not perform pairwise comparisons during inference to obtain the predicted rank assignments due to the high time complexity (this high complexity is avoided during training due to training list truncation, see Appendix B.2). However, our theory is applicable to any (black-box) model that produces list level representations, and our ListDA method is also more generally applicable any models whose list level representations are stackings of feature vectors, i.e., $z = (u_1, u_2, \cdots, u_L)$ and $u_L \in \mathbb{R}^k$ (here $L \neq \ell$, and can be arbitrary). For instance, while not pursued in this work, ListDA could be instantiated on DuoT5 (Pradeep et al., 2021), a SOTA pairwise ranking model, by treating the stackings of pairwise $q$-$d_i$-$d_j$ embeddings as the list feature representations (thereby $L = \ell(\ell - 1)/2$).

**ListDA + QGen PL Method and Signal-1M Dataset.**   We experiment with supplementing ListDA with QGen pseudolabels by (uniformly) combining the training objectives of ListDA and QGen PL methods (**ListDA + QGen PL**). The results on the three datasets considered in Section 5 are included in Table 2b. This method is also applied on the Signal-1M (RT) dataset (Suarez et al., 2018), with results in Table 3. It is noted that reranking using neural rerankers transferred from MS MARCO source domain do not give better performance than BM25 on Signal-1M, which is also observed in prior work (Thakur et al., 2021; Liang et al., 2022). This does not mean that neural rerankers are worse than BM25, but that MS MARCO is not a good choice for source domain when Signal-1M is the target, because of the arguably large domain shift between tweet retrieval and MS MARCO web search—qualitatively, it can be seen from Table 7 that the text styles and task semantics of the two domains are very different. Hence, the following discussions on Signal-1M results are focused on reranking models.

On Signal-1M, QGen PL improves upon the zero-shot baseline but ListDA does not, which is likely due to the large domain shift between MS MARCO and Signal-1M that prevented ListDA from finding the correct source-target feature alignment without supervision. With QGen pseudolabels, improved ListDA performance is observed with + QGen PL, which could have benefited from QGen

Table 2: Transfer performance of T5 reranker on top 1000 BM25-retrieved passages; in addition to Table 1 results.

(a) With pairwise logistic ranking loss in place of softmax cross-entropy on Robust04.

| Target Domain | Method | MAP | MRR@10 | NDCG@5 | NDCG@10 | NDCG@20 |
|---|---|---|---|---|---|---|
| Robust04 | Zero-shot | 0.2656 | 0.7894 | 0.5671 | 0.5163 | 0.4729 |
| | QGen PL | 0.2776* | 0.7975 | 0.5576 | 0.5267 | 0.4892* |
| | ItemDA | 0.2766* | 0.8021 | 0.5794 | 0.5355* | 0.4917* |
| | ListDA | **0.2893**\*†‡ | **0.8103** | **0.5935**\*†‡ | **0.5524**\*†‡ | **0.5044**\*†‡ |

Source domain is MS MARCO. Gain function in NDCG is the identity map. *Improves upon zero-shot baseline with statistical significance ($p \leq 0.05$) under the two-tailed Student's $t$-test. †Improves upon QGen PL. ‡Improves upon ItemDA.

(b) With ListDA + QGen PL domain adaptation method.

| Target Domain | Method | MAP | MRR@10 | NDCG@5 | NDCG@10 | NDCG@20 |
|---|---|---|---|---|---|---|
| Robust04 | | 0.2851*† | 0.8039† | 0.5761† | 0.5386† | 0.4975† |
| TREC-COVID | ListDA + QGen PL | 0.3168 | 0.8950 | 0.8539 | 0.8292 | 0.7820 |
| BioASQ | | 0.6538*‡ | 0.5158 | 0.5547‡ | 0.5671*‡ | 0.5931*‡ |

Source domain is MS MARCO. Gain function in NDCG is the identity map. *Improves upon zero-shot baseline with statistical significance ($p \leq 0.05$) under the two-tailed Student's $t$-test. †Improves upon QGen PL. ‡Improves upon ItemDA.

Table 3: Transfer performance of T5 reranker on top 1000 BM25-retrieved passages on Signal-1M.

| Target Domain | Method | MAP | MRR@10 | NDCG@5 | NDCG@10 | NDCG@20 |
|---|---|---|---|---|---|---|
| Signal-1M | BM25 | **0.1740** | **0.5765** | **0.3639** | **0.3215** | **0.2905** |
| | Zero-shot | 0.1511 | 0.4804 | 0.3068 | 0.2685 | 0.2410 |
| | QGen PL | 0.1541 | 0.5043 | 0.3238 | 0.2799 | 0.2497 |
| | ListDA | 0.1456 | 0.4629 | 0.3002 | 0.2602 | 0.2328 |
| | ListDA + QGen PL | 0.1549 | 0.5170 | 0.3261 | 0.2817 | 0.2505 |

Source domain is MS MARCO. Gain function in NDCG is the identity map.

q-d pairs acting as anchor points for ListDA to find the correct correspondence between source and target.

Overall, ListDA + QGen PL is the only method that consistently improves upon the zero-shot baseline on all four datasets including Signal-1M, although underperforms ListDA on the other three. Further improvements to this method may be possible with better strategies of balancing the constituent training objectives of ListDA and QGen PL.

**Case Studies.** In Tables 8 and 9, we include examples where the ranks (top-1) predicted by models trained with ListDA have higher utilities v.s. zero-shot and QGen PL results, respectively, to provide a qualitative understanding of the benefits and advantages of ListDA.

In zero-shot learning, the reranker is trained on MS MARCO general domain data only and directly evaluated on the target domains. When the target domain differs from MS MARCO stylistically or consists of passages containing domain-specific words, as in the TREC-COVID and BioASQ datasets for biomedical retrieval, the zero-shot model, which is barely exposed to the specialized language usages, may resort to keyword matching. Examples of such cases are presented in Table 8, where the top passages returned by the zero-shot model contain keywords from the query but are irrelevant.

In QGen PL, the reranker is trained on the pseudolabels generated during the query synthesis procedure, treating the passages from which the queries are generated as relevant. However, as remarked in Section 5, because QGen is deployed in a zero-shot manner, the pseudolabels are not guaranteed to be valid, meaning that they could be false positives. This is observed in cases presented in Tables 7 and 9. One specific scenario where false positives hurt transfer performance is when the synthesized queries (of false positive documents) coincide with queries from the evaluation set, which is indeed

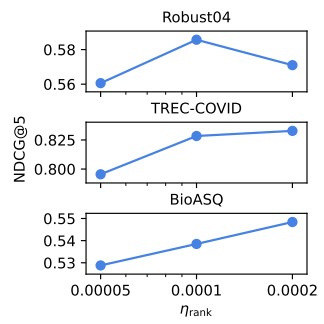
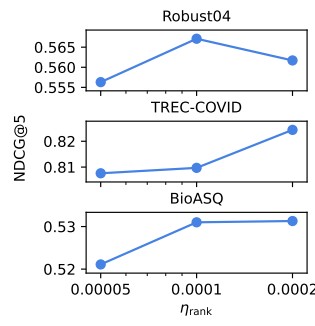

(a) With softmax cross-entropy ranking loss.      (b) With pairwise logistic ranking loss.

Figure 4: Zero-shot performance under different hyperparameter settings for $\eta_{\text{rank}}$.

the case on TREC-COVID and BioASQ in Table 9! Since ListDA does not assume the pseudolabels in its training objective, it does not suffer the same pitfalls as QGen PL.

### B.1 HYPERPARAMETER SETTINGS

We discuss model hyperparameter settings for our passage reranking experiments in this section.

For BM25, we use the implementation of Anserini (Yang et al., 2017), set $k_1 = 0.82$ and $b = 0.68$ on MS MARCO source domain, and $k_1 = 0.9$ and $b = 0.4$ on all target domains without tuning. As in (Thakur et al., 2021), titles are indexed as a separate field with equal weight as the document body, if available.

For the T5 reranker, the model is fine-tuned from the T5 v1.1 Base checkpoint on a Dragonfish TPU with 8x8 topology for 100,000 steps with batch size of 32 (each example is a list containing 31 items) per domain. We tune the learning rate $\eta_{\text{rank}} \in \{5e-5, 1e-4, 2e-4\}$, and select the one that gives the best zero-shot performance to use on all models for each dataset (see Fig. 4 for zero-shot sweep results). We apply a learning rate schedule on $\eta_{\text{rank}}$ that decays (exponentially) by a factor of 0.7 every 5,000 steps. Each concatenated query-document text input are truncated to 512 tokens.

For the domain discriminators, there are two hyperparameters: the strength of invariant feature learning $\lambda$, and the discriminator learning rate $\eta_{\text{ad}}$. We tune both by sweeping $\lambda \in \{0.01, 0.02\}$, and $\eta_{\text{ad}} \in \{10, 20, 40\} \times \eta_{\text{rank}}$, multiples of the reranker learning rate (see Fig. 3 in Section 5 for ListDA sweep results). The tuned hyperparameter settings for $\eta_{\text{rank}}$, $\eta_{\text{ad}}$ and $\lambda$ used in our experiments are included in Table 4.

As a remark on running time, ListDA and ItemDA take roughly the same time to train as QGen PL because the overhead of the domain discriminators is small. Compared to zero-shot, they have double the training time due to data loading: the adaptation methods are trained on target domain data in addition to source domain ones.

### B.2 TRAINING EXAMPLE LIST CONSTRUCTION

Recall from the description in Section 5 that in listwise ranking, the inputs are defined over lists and the invariant representations for domain adaptation are learned at the list level. In other words, the ranking loss and the adversarial loss need to be computed on ranking scores and feature representations that the model outputs on lists of documents (containing both relevant and irrelevant ones) for each query.

Under our reranking setup, each list would be the top-$r$ documents retrieved by BM25 on a query, and we set $r = 1000$ in our experiments. However, there are two complications. The first is that due to memory constraints, it is not feasible to feed all 1000 documents from each list simultaneously through the T5 encoder during training. The second is that the source domain MS MARCO dataset only contains annotations of one relevant document per query, meaning that out of the 1000

Table 4: Hyperparameter settings of T5 reranker and domain discriminators.

(a) With softmax cross-entropy ranking loss.

| Target Domain | Method | $\eta_{\text{rank}}$ | $\eta_{\text{ad}}$ | $\lambda$ |
|---|---|---|---|---|
| Robust04 | Zero-shot | | - | - |
| | QGen PL | | - | - |
| | ItemDA | 1e-4 | 1e-3 | 0.01 |
| | ListDA | | 4e-3 | 0.01 |
| | ListDA + QGen PL | | 1e-3 | 0.02 |
| TREC-COVID | Zero-shot | | - | - |
| | QGen PL | | - | - |
| | ItemDA | 2e-4 | 8e-3 | 0.01 |
| | ListDA | | 4e-3 | 0.01 |
| | ListDA + QGen PL | | 4e-3 | 0.02 |
| BioASQ | Zero-shot | | - | - |
| | QGen PL | | - | - |
| | ItemDA | 2e-4 | 4e-3 | 0.01 |
| | ListDA | | 8e-3 | 0.01 |
| | ListDA + QGen PL | | 4e-3 | 0.02 |
| Signal-1M | Zero-shot | | - | - |
| | QGen PL | | - | - |
| | ListDA | 5e-5 | 2e-3 | 0.01 |
| | ListDA + QGen PL | | 1e-3 | 0.02 |

(b) With pairwise logistic ranking loss.

| Target Domain | Method | $\eta_{\text{rank}}$ | $\eta_{\text{ad}}$ | $\lambda$ |
|---|---|---|---|---|
| Robust04 | Zero-shot | | - | - |
| | QGen PL | | - | - |
| | ItemDA | 1e-4 | 2e-3 | 0.01 |
| | ListDA | | 2e-3 | 0.01 |

documents retrieved by BM25 for each query, we would only know that one of them is relevant; the ground-truth relevance scores for the remaining 999 documents is unknown.

**Example List Construction with Negative Sampling.** To address both issues, we truncate the list to length $\ell = 31$ during training, and perform random negative sampling from the BM25 results to gather irrelevant documents.

On the MS MARCO source domain, given a query $q$ and top 1000 documents retrieved by BM25, $d_1, \cdots, d_{1000}$, we construct the example list with the one document $d^*$ labeled as relevant in the dataset along with 30 randomly sampled documents $d_{N_1}, \cdots, d_{N_{30}}$ treated as irrelevant, and get $x = ([q, d^*], [q, d_{N_1}], \cdots, [q, d_{N_{30}}])$ and $y = (1, 0, \cdots, 0)$.

For the target domains, we perform the same procedure. Given a QGen synthesized query and top 1000 documents retrieved by BM25, the example list consists of the pseudolabeled document $\hat{d}$ (i.e., the document with which the query was synthesized) and 30 randomly sampled irrelevant documents, so that $x = ([q, \hat{d}], [q, d_{N_1}], \cdots, [q, d_{N_{30}}])$ and $y = (1, 0, \cdots, 0)$. Note that the pseudolabels $y$ are used by QGen PL but discarded by ListDA.

**Reducing MS MARCO False Negatives.** One potential problem that arises with negative sampling is that the constructed lists may contain false negatives (i.e., relevant documents that are incorrectly marked as irrelevant); in fact, false negatives are prevalent in the MS MARCO dataset. While such false negatives are relatively harmless for source domain supervised training because the true positive document to which they are similar is labeled and trained on so that the effects are canceled out, they have a larger impact on source-target feature alignment for invariant representation learning.

One potential negative impact is that the duplicates in the lists (which have identical feature vectors) will cause ListDA to collapse distinct documents on the target domain to the same feature vector for achieving alignment, which is an artifact that will cause information loss in target domain feature

Table 5: Transfer performance of 3-layer MLP ranker on Yahoo! LETOR (Set 2).

| Target Domain | Method | MAP | MRR@10 | NDCG@5 | NDCG@10 | NDCG@20 |
|---|---|---|---|---|---|---|
| Yahoo! LETOR (Set 2) | Zero-shot | 0.5138 | 0.6558 | 0.7302 | 0.7627 | 0.8199 |
| | Supervised | 0.5389 | 0.6785 | 0.7523 | 0.7829 | 0.8341 |
| | ItemDA | 0.5315* | 0.6717* | 0.7402* | 0.7708* | 0.8255* |
| | ListDA | **0.5370**\*‡ | **0.6771**\*‡ | **0.7442**\*‡ | **0.7735**\*‡ | **0.8269**\*‡ |

Source domain is Yahoo! LETOR (Set 1). Gain function in NDCG is the identity map. Bold indicates the best unsupervised result. Results are from ensembles of five models. *Improves upon zero-shot baseline with statistical significance ($p \leq 0.05$) under the two-tailed Student's $t$-test. ‡Improves upon ItemDA. Significance tests are not performed on supervised results.

representations. Another is that the inclusion of duplicates will alter the marginal distribution of scores on the source domain(Zhao et al., 2019).

To lower the chance of selecting false negatives on MS MARCO, we rerank each BM25-retrieved results using a ranker that is pre-trained on MS MARCO, and sample negatives from documents that are ranked at 300 or higher, since the duplicates and relevant documents will be concentrated at the top (Qu et al., 2021). We only apply this method when constructing source domain lists for feature alignment (namely, in ListDA and ItemDA), and it only affects the computation of adversarial loss. This sampling method is not applicable to target domains because we do not have reliable pre-trained rankers for them. It is also not used for source domain supervised training (i.e., the computation of ranking loss), as decreased performance was observed in our preliminary experiments with this method, likely due to the exclusion of "hard" negatives.

## C  EXPERIMENTS ON YAHOO! LETOR DATASET

Our method is also evaluated on the ranking task from the Yahoo! Learning to Rank Challenge v2.0 (Chapelle & Chang, 2011), referred to as the Yahoo! LETOR dataset. This is a web search ranking dataset in numerical format, where each item is represented by a 700-d vector with values in the range of $[0, 1]$. It has two subsets, called "Set 1" and "Set 2", whose data originate from the US and an Asian country respectively. Among the 700 features, 415 are defined on both sets (shared), and the other 285 are defined only on Set 1 or 2 only (disjoint); we hence write each item $x := [x_{\text{shared}}, x_{\text{disjoint}}]$ as a concatenation of shared features $x_{\text{shared}} \in \mathbb{R}^{415}$ and disjoint ones $x_{\text{disjoint}} \in \mathbb{R}^{285}$.

We consider unsupervised domain adaptation from Set 1 to Set 2. Our code is implemented with the Hugging Face Transformers library (Wolf et al., 2020).

**Models.**  Our models has the same setup as that of the passage reranking experiments in Section 5, except that the T5 reranker is replaced by a three-hidden-layer MLP following (Zhuang et al., 2020), and we treat the list of 256-d outputs on the second layer as the feature representations.

$$g(x)_i = u_i = \text{ReLU}\bigg( W_3 \begin{bmatrix} \text{ReLU}(W_2\text{ReLU}(W_1 x_{i,\text{shared}} + b_1) + b_2) \\ \text{ReLU}(W_2'\text{ReLU}(W_1' x_{i,\text{disjoint}} + b_1') + b_2') \end{bmatrix} + b_3 \bigg),$$
$$h(x)_i = s_i = W_4 u_i + b_4,$$

where $W_1 \in \mathbb{R}^{1024\times415}$, $W_1' \in \mathbb{R}^{1024\times285}$, $W_2, W_2' \in \mathbb{R}^{256\times1024}$, $W_3 \in \mathbb{R}^{256\times512}$, and $W_4 \in \mathbb{R}^{1\times256}$, all randomly initialized.

Each model in the ensemble of five domain discriminators is a stack of three T5 encoder transformer blocks, with 4 attention heads (`num_heads`), size-32 key, query and value projections per attention head (`d_kv`), and size-1024 intermediate feedforward layers (`d_ff`).

**Results.**  The results are presented in Table 5. Considering the small dataset size and number of training steps, each method is evaluated by an ensemble of five separately trained models to reduce the variance on the results due to the randomness in the initialization and the training process. Since Yahoo! LETOR is annotated with 5-level relevancy, the scores are binarized for MAP and MRR metrics by mapping 0 (bad) and 1 (fair) to negative, and 2 (good), 3 (excellent), 4 (perfect) to

Table 6: Hyperparameter settings of 3-layer MLP ranker and domain discriminators.

| Target Domain | Method | $\eta_{rank}$ | $\eta_{ad}$ | $\lambda$ |
|---|---|---|---|---|
| Yahoo! LETOR (Set 2) | Zero-shot | 4e-4 | - | - |
| | Supervised | 4e-5 | - | - |
| | ItemDA | 2e-4 | 4e-4 | 0.4 |
| | ListDA | 2e-4 | 1.6e-3 | 0.1 |

positive. Because of the availability of labeled data on Set 2, we also include **Supervised** results, where the model is trained on labeled data from both Set 1 and 2.

The best unsupervised transfer performance is achieved by ListDA. In particular, the favorable comparison of ListDA to ItemDA again corroborates our discussion in Section 3 that listwise invariant representation learning is more appropriate for listwise ranking, although their gap is smaller compared to passage reranking results (Table 1), which we suspect is because the weak contextual (query) information for defining the list structure on the data is too weak in this numerical dataset. Lastly, thanks to the availability of labeled data, we also provide **Supervised** results where the model is trained on both Set 1 and 2, serving as an upper bound for ListDA.

**Hyperparameters.** The model is trained from scratch on an NVIDIA A6000 GPU for 5,000 steps with batch size of 32 (each example is a list containing one to no more than 140 items) per domain. We apply a learning rate schedule on $\eta_{rank}$ that decays (exponentially) by a factor of 0.7 every 500 steps.

We exhaustively tune ranker and domain discriminator learning rates and the strength of feature alignment from $\eta_{rank} \in \{$1e-5, 2e-5, 4e-5, 8e-5, 1e-4, 2e-4, 4e-4, 8e-4, 1e-3$\}$, $\eta_{ad} \in \{$0.2, 0.4, 0.8, 1, 2, 4, 8, 10$\} \times \eta_{rank}$, and $\lambda \in \{$0.01, 0.02, 0.04, 0.08, 0.1, 0.2$\}$. The tuned settings are included in Table 6.

Table 7: Samples of test set relevant q-d pairs and QGen synthesized q-d pairs from domains considered in Section 5 passage reranking experiments. Truncated or omitted texts are indicated by "[...]".

| Dataset | Ground-Truth Relevant Q-D Pairs | QGen Q-D Pairs |
|---------|--------------------------------|----------------|
| MS MARCO | D: What is cartography? A. the science of mapmaking B. the science of shipbuilding C. the science of charting direction on a ship D. the science of measuring distances on the ocean. Cartography is the science of map making A.

Q: what is the science of mapmaking called

D: The flu shot also contains the following ingredients: sodium phosphate & buffered isotonic sodium chloride solution, formaldehyde, octylphenol ethoxylate, and gelatin, according to the FDA.

Q: what's in the flu shot | - |
| TREC-COVID | D: An Evidence Based Perspective on mRNA-SARS-CoV-2 Vaccine Development. [...] The production of mRNA-based vaccines is a promising recent development in the production of vaccines. However, there remain significant challenges in the development [...]

Q: what is known about an mRNA vaccine for the SARS-CoV-2 virus?

D: The possible pathophysiology mechanism of cytokine storm in elderly adults with COVID-19 infection: the contribution of "inflame-aging". PURPOSE: Novel Coronavirus disease 2019 (COVID-19), is an acute respiratory distress syndrome (ARDS), [...]

Q: What is the mechanism of cytokine storm syndrome on the COVID-19? | D: Impact of arterial load on the agreement between pulse pressure analysis and esophageal Doppler. INTRODUCTION. The reliability of pulse pressure analysis to estimate cardiac output is known to be affected by arterial load changes. [...]

QGen: what is arterial load for pulse pressure analysis

D: Opportunity Costs Pacifism. If the resources used to wage wars could be spent elsewhere and save more lives, does this mean that wars are unjustified? This article considers this question, which has been largely overlooked by Just War Theorists and pacifists. It focuses on whether the opportunity costs of war [...]

QGen: opportunity cost pacifism |
| BioASQ | D: The role of extended-release amantadine for the treatment of dyskinesia in Parkinson's disease patients. [...] Extended-release amantadine (amantadine ER) is the first approved medication for the treatment of dyskinesia. When it is given at bedtime, it [...]

Q: Is amantadine ER the first approved treatment for akinesia?

D: [...] We investigated the health-related quality of life (HRQoL) of long-term prostate cancer patients who received leuprorelin acetate in microcapsules (LAM) for androgen-deprivation therapy (ADT). [...]

Q: Can leuprorelin acetate be used as androgen deprivation therapy? | D: Subluxation of the femoral head in coxa plana. Twenty-two patients who had severe coxa plana had closed reduction for lateral subluxation of the femoral head, [...] The average age when the patients were first seen was eight years and six months. [...]

QGen: average age of femoral subluxation

D: [...] a comparison of proxy assessment and patient self-rating using the disease-specific Huntington's disease health-related quality of life questionnaire (HDQoL). [...] Specific Scales of the HDQoL. On the Specific Hopes and Worries Scale, proxies on average rated HrQoL as better than patients' [...]

QGen: which scale is used for proxy assessment of hrqol |
| Signal-1M | D: BJP terms party MP R.K Singh's allegation that money has changed hands for tickets in #BiharPolls as baseless.

Q: Party MP calls BJP 'Baura Jayewala Party'

D: Kerry: US plans military talks with Russia over Syria

Q: Kerry: US plans military talks with Russia over Syria | D: Black lives matter: thoughts from the delivery ward in St. Louis: #mustread

QGen: where is black lives matter?

D: RETWEET if "Brenda's Got A Baby" is one of your favorite @2Pac songs. #RIP2Pac

QGen: brenda got a baby pac |

Table 8: Samples of passage reranking results where ListDA achieves higher utilities v.s. zero-shot. Truncated or omitted texts are indicated by "[...]".

| Dataset | Zero-Shot Top Results (Ground-Truth Irrelevant) | ListDA Top Results (Ground-Truth Relevant) |
|---|---|---|
| Robust04 | Q: Find information on prostate cancer detection and treatment. | |
| | D: [...] FIRST PATIENT UNDERGOES GENE INSERTION IN CANCER TREATMENT [...] This first round of gene transfer experiments, in which a gene was inserted into a patient's white blood cells, is not expected to directly benefit an individual patient. Instead, the inserted gene is being used to track the movement in the body of the cancer-fighting white blood cells. [...] Inserting human genes to repair defects may one day help with a host of inherited disorders, [...] | D: [...] Little knowledge goes a long way - Cancer of the prostate need not be a killer / Health Check. Earlier this year, 13-year-old [...] died from cancer of the bladder and prostate. His death is a grim reminder that no male should consider himself immune from water-works trouble. The prostate, a gland about the size and shape of a chestnut, lies deep in the pelvis just below the bladder. Because it surrounds the urethra, it has the potential to block the flow of urine completely. [...] |
| TREC-COVID | Q: What are the longer-term complications of those who recover from COVID-19? | |
| | D: [...] Our previous experience with members of the same corona virus family (SARS and MERS) which have caused two major epidemics in the past albeit of much lower magnitude, has taught us that the harmful effect of such outbreaks are not limited to acute complications alone. Long term cardiopulmonary, glucometabolic and neuropsychiatric complications have been documented following these infections. [...] | D: Up to 20-30% of patients hospitalized with coronavirus disease (COVID-19) have evidence of myocardial involvement. Acute cardiac injury in patients hospitalized with COVID-19 is associated with higher morbidity and mortality. There are no data on how acute treatment for COVID-19 may affect convalescent phase or long-term cardiac recovery and function. Myocarditis from other viral pathogens can evolve into overt or subclinical myocardial dysfunction, [...] |
| BioASQ | Q: What is the interaction between WAPL and PDS5 proteins? | |
| | D: Pds5 and Wpl1 act as anti-establishment factors preventing sister-chromatid cohesion until counteracted in S-phase by the cohesin acetyl-transferase Eso1. [...] Here, we show that Pds5 is essential for cohesin acetylation by Eso1 and ensures the maintenance of cohesion by promoting a stable cohesin interaction with replicated chromosomes. The latter requires Eso1 only in the presence of Wapl, indicating that cohesin stabilization relies on Eso1 only to neutralize the anti-establishment activity. [...] | D: [...] Here, we show that cohesin suppresses compartments but is required for TADs and loops, that CTCF defines their boundaries, and that the cohesin unloading factor WAPL and its PDS5 binding partners control the length of loops. In the absence of WAPL and PDS5 proteins, cohesin forms extended loops, presumably by passing CTCF sites, accumulates in axial chromosomal positions (vermicelli), and condenses chromosomes. [...] |

Table 9: Samples of passage reranking results where ListDA achieves higher utilities v.s. QGen PL. Truncated or omitted texts are indicated by "[...]".

| Dataset | QGen PL Top Results (Ground-Truth Irrelevant) | ListDA Top Results (Ground-Truth Relevant) |
|---|---|---|
| Robust04 | Q: Identify outbreaks of Legionnaires' disease. | |
| | D: [...] 3. Care of Patients with Tracheostomy 4. Suctioning of Respiratory Tract Secretions III. Modifying Host Risk for Infection A. Precautions for Prevention of Endogenous Pneumonia 1. Prevention of Aspiration 2. Prevention of Gastric Colonization B. Prevention of Postoperative Pneumonia C. Other Prophylactic Procedures for Pneumonia 1. Vaccination of Patients 2. Systemic Antimicrobial Prophylaxis 3. Use of Rotating "Kinetic" Beds Prevention and Control of Legionnaires' Disease [...]

QGen: what kind of precautions are used to prevent pneumonia | D: [...] LEGIONNAIRE'S DISEASE STRIKES 16 AT REUNION IN COLORADO; 3 DIE. An outbreak of legionnaire's disease at a 50th high school reunion was blamed Thursday for the deaths of three elderly celebrants and the pneumonia-like illness of 13 others. State health officials contacted 250 other people from 21 states who attended the Lamar High School reunion for the classes of 1937 through 1941 but found no new cases, Dr. Ellen Mangione, a Colorado Department of Health epidemiologist, said. [...] |
| TREC-COVID | Q: what drugs have been active against SARS-CoV or SARS-CoV-2 in animal studies? | |
| | D: Different treatments are currently used for clinical management of SARS-CoV-2 infection, but little is known about their efficacy yet. Here we present ongoing results to compare currently available drugs for a variety of diseases to find out if they counteract SARS-CoV-2-induced cytopathic effect in vitro. [...] We will provide results as soon as they become available, [...]

QGen: what is the treatment for sars | D: [...] the antiviral efficacies of lopinavir-ritonavir, hydroxychloroquine sulfate, and emtricitabine-tenofovir for SARS-CoV-2 infection were assessed in the ferret infection model. [...] all antiviral drugs tested marginally reduced the overall clinical scores of infected ferrets but did not significantly affect in vivo virus titers. Despite the potential discrepancy of drug efficacies between animals and humans, these preclinical ferret data should be highly informative to future therapeutic treatment of COVID-19 patients. |
| BioASQ | Q: What is the function of the Spt6 gene in yeast? | |
| | D: As a means to study surface proteins involved in the yeast to hypha transition, human monoclonal antibody fragments (single-chain variable fragments, scFv) have been generated that bind to antigens expressed on the surface of Candida albicans yeast and/or hyphae. [...] To assess C. albicans SPT6 function, expression of the C. albicans gene was induced in a defined S. cerevisiaespt6 mutant. Partial complementation was seen, confirming that the C. albicans and S. cerevisiae genes are functionally related in these species.

QGen: what is the function of spt6 gene in candida albicans | D: Spt6 is a highly conserved histone chaperone that interacts directly with both RNA polymerase II and histones to regulate gene expression. [...] Our results demonstrate dramatic changes to transcription and chromatin structure in the mutant, including elevated antisense transcripts at >70% of all genes and general loss of the +1 nucleosome. Furthermore, Spt6 is required for marks associated with active transcription, including trimethylation of histone H3 on lysine 4, previously observed in humans but not Saccharomyces cerevisiae, and lysine 36. [...] |

