# OpenReview forum: "Learning Listwise Domain-Invariant Representations for Ranking"
_ICLR.cc/2023/Conference — Submitted to ICLR 2023_

### Official Review · Reviewer_HnLS · 2022-10-24

**Confidence:** 4
**Correctness:** 4
**Technical Novelty And Significance:** 3
**Empirical Novelty And Significance:** 2
**Recommendation:** 5

**Clarity, Quality, Novelty And Reproducibility:**

This paper provides concrete mathematical proofs to explain the insights and illustrative diagrams to elaborate the procedure of text ranking with the proposed method. In general, the writing of this paper is clear. However, more concrete examples can be given, such as what kind of passages for which the proposed method performs better or less satisfying.

Extensive experiments have been conducted and the presented results are mostly convincing.

It's theoretically novel to tailor domain-invariant representation learning to the listwise approach for learning to rank. However, the contribution of this paper is limited to the problem of reRanking candidate documents, rather than the fundamental ranking problem in general.

**Strength And Weaknesses:**

Strength:

1. This paper is theoretically sound, providing an important insight that when the domain shift is small according to the Wasserstein distance, a trained ranking model can be domain-adaptable, and its performance under regular ranking metrics can be bounded.

2. The experimental results are reasonable and the proposed ListDA methods outperform baselines significantly by minimizing the source and target distributional shifts for learning domain-invariant representations.

Weakness:

1. Although the comparison between learning domain-invariant representation in the listwise and pointwise settings is given, the connection to the pairwise ranking has been little discussed.

2. The evaluation is on the reRanking problem, that is reranking a list of candidate documents retrieved by a first-stage retrieval model in response to a search query, rather than the fundamental ranking problem. This seems to be one of the main limitations of this paper. The discussion about why the proposed method is not applied to the fundamental ranking problem can be elaborated.


**Summary Of The Paper:**

This paper introduces a method to learn invariant listwise representations for ranking. A theoretical generalization bound is presented by analyzing domain adaptation for learning to listwise rank. Experimental results demonstrate the effectiveness of the proposed method on unsupervised domain adaptation for passage reRanking on a diversity of domains including biomedical and news articles.

**Summary Of The Review:**

An adversarial training method is introduced in this paper for learning domain-invariant representations to generalize the ranking model from the source to the target domain, in the setting of listwise ranking. Experiments on passage reRanking demonstrate the superiority of the proposed method.

The paper is clear and illustrative with concrete proofs and diagrams.

However, the application and evaluation of this paper are limited to the reRanking problem. The contribution of this paper can be enhanced if the authors could generalize the proposed method to the fundamental ranking problem.

Also, along with listwise ranking and pointwise ranking, it'd be great if the authors could discuss more about pairwise ranking. It'd be also interesting to compare with some of the SOTA pairwise ranking methods.

Presenting a complexity analysis of the proposed method and comparison to baselines is recommended, in order to show the efficiency of ListDA.

---

> ### Author Response · Authors · 2022-11-18
> **Response to Reviewer HnLS (Part 1 of 2)**
>
> Thank you for your helpful comments!  And we hope that the following addresses the concerns in the review:
>
> -   **The connection [of learning domain-invariant representation] to the pairwise ranking has been little discussed...  It'd be also interesting to compare with some of the SOTA pairwise ranking methods.**
>
>     Our theory suggests that a bound on target performance under listwise ranking metrics (e.g., MRR and NDCG) can be achieved via learning listwise invariant representations.  Therefore, as long as the model (feature map) can be treated as a black box that outputs feature representations on a list level, our theory and method (Sections 3 and 4) can be applied, regardless if it is a listwise or pairwise model.
>
>     Between listwise (our experiments) and pairwise ranking, the two main differences are that the ranking loss is computed on all pairs in the list, and during inference, the predicted rank assignments are assembled from pairwise comparisons.  Therefore, to instantiate our adaptation method on pairwise ranking, we only need to swap the training ranking loss and the inference procedure; the domain discriminator and adversarial training components can be directly borrowed from our listwise instantiation.
>
>     We have added a new set of experiments in Table 2(a) on page 19 using the pairwise logistic ranking loss on the Robust04 dataset, and the findings from the results are consistent with earlier ones.  Due to time constraints, we did not perform experiments using SOTA pairwise models e.g. DuoT5 [5] (where the list representation would have length $\ell(\ell-1)/2$).  This discussion is added following the experiments with pairwise ranking loss.
>
>     Indeed, a comprehensive study/survey of domain adaptation for ranking would also include SOTA pairwise models, but is out of the scope of our work focused on establishing an initial theoretical foundation for domain adaptation for ranking problems.  The current set of experiments in our paper has demonstrated our message that learning representations at the list level is more appropriate than at the item level under listwise metrics, and it is our hope that our results could inspire more work in this arguably underexplored area.
>
> -   **The evaluation is on the reRanking problem...  This seems to be one of the main limitations of this paper.**
>
>     Thank you for the remark!  Conclusions drawn from our theory for domain adaptation (Theorem 2 and Section 3) are applicable to all learning to rank models/systems that involve feature learning, including reranking and retrieval (with e.g. dual encoders)—both fundamental research problems in information retrieval.  Due to space constraints, experiment-wise, the paper focuses on the reranking task, but the methods are also applicable to retrieval models.  Some existing work already tackles domain adaptation for retrievers, e.g., [4] applies invariant representation learning to adapt dual encoders for dense retrieval.  However, [4] learns invariant representations on the item-level and not the list-level, the latter of which is demonstrated in our work to be more appropriate under listwise metrics.  Applying listwise invariant representation learning to unsupervised domain adaptation for retrievers still has its own non-trivial challenges, e.g, how to construct the list representatively and efficiently for each query, and what is the most appropriate way to derive the list representation.  We will leave these as future work.
>
>     We also have performed a new set of domain adaptation experiments on the Yahoo! Learning to Rank Challenge v2.0 dataset, a web search ranking task on numerical data.  The lists to be ranked are directly provided by the data set, so the experiment setup is potentially a more fundamental ranking problem.  The experiment details are provided in Appendix C of the revised draft, and the results are consistent with those from the passage reranking experiments and also support our earlier findings.  We exhaustively tuned the hyperparameters for all methods on this dataset, and plan to release the code on the Yahoo! LETOR experiments after the anonymity period for the purpose of reproducibility.

---

> > ### Author Response · Authors · 2022-11-18
> > **Response to Reviewer HnLS (Part 2 of 2)**
> >
> > -   **More concrete examples can be given, such as what kind of passages for which the proposed method performs better or less satisfying.**
> >
> >     Thank you for the suggestion: we have added case studies in Tables 8 and 9 along with discussions in Appendix B (page 19) in the revision.  In the comparison between our proposed method (ListDA) and the baseline of QGen PL, we observed cases where QGen may synthesize a query to which the input passage is in fact not relevant (i.e., false positive), but the query (interestingly) happens to coincide with real-world queries in the evaluation set!  Since QGen PL trains on the pseudolabels, including false positives, this will cause the model to return irrelevant documents that were mistakenly pseudolabeled as positive on such queries.  ListDA, on the other hand, avoids the same pitfalls as it does not assume the pseudolabels in its training objective.  We also observed that zero-shot may resort to keyword matching when the query or document contains specialized usage of language, e.g., in biomedical ranking.
> >
> > -   **Presenting a complexity analysis of the proposed method... to show the efficiency of ListDA.**
> >
> >     In terms of running time, ListDA and ItemDA take roughly the same time to train as QGen PL because the overhead of the domain discriminators is small.  Compared to zero-shot, they have double the training time due to data loading: the adaptation methods are trained on target domain data in addition to source domain ones.
> >
> >     In terms of data efficiency, we plot the empirical performance of ListDA under varying target data sizes in Fig. 2 (with discussions in Section 5.2).  Theoretically, sample complexity bound for ListDA could be derived via standard complexity analyses from our Theorem 2, as suggested in the paper, and similar results have appeared in prior work [1, 2].  We did not pursue this direction, however, because of the limited utility of such results in unsupervised domain adaptation settings due to the unknown term λ* (optimal joint risk).  Ideally, learning bounds would be preferred—out of the scope of this work—which are currently actively studied under additional assumptions [3].
> >
> > [1]  Analysis of Representations for Domain Adaptation. 2007.
> > [2]  Learning Bounds for Domain Adaptation. 2008.
> > [3]  Connect, Not Collapse: Explaining Contrastive Learning for Unsupervised Domain Adaptation. 2022.
> > [4]  Zero-Shot Dense Retrieval with Momentum Adversarial Domain Invariant Representations. 2022.
> > [5]  The Expando-Mono-Duo Design Pattern for Text Ranking with Pretrained Sequence-to-Sequence Models. 2021.

---

### Official Review · Reviewer_U7gj · 2022-10-25

**Confidence:** 3
**Correctness:** 4
**Technical Novelty And Significance:** 3
**Empirical Novelty And Significance:** 3
**Recommendation:** 8

**Clarity, Quality, Novelty And Reproducibility:**

Paper is fairly clear and novelty is incremental.
I may have missed it, but not sure if the results are reproducible with their code, they do give hyperparameter details.

**Strength And Weaknesses:**

Strengths:
Paper is fairly self contained
Paper shows both theoretical bounds and practical approach.

Weakness:
The writing in approach section blends both previous work and their approach making a bit hard to discern their contribution.
The experimental set-up might have some issues:
* Why was the better negative sampling not used for other approaches.
* While it is true that the paper focuses on ranking and hence retrieval model can be basic, it is unclear how their approach might be affected by a better retriever that could be encoder-based.
* The model training is a bit confusing, should the adversarial training be done for every new target distribution or can it be trained once and re-used?
* Q-D generation is a bit questionable, if the source target dont have many documents from the new domain would it still be ok? what is the threshold that determines that? Would be good if authors mention some details.

Minor comments:
* There are assumptions made in section 3 for the proof, but not all assumptions are justified. Would be good if they can add a couple of lines on why they are reasonable.
* There are some minor grammatical errors which sometimes make it hard to understand like: "The final missing piece is the choice of $F_{ad}$ that can model lists $z = (z1 , · · · , zl )$ of feature vectors zi ∈ Rd and is continuously differentiable."
* It would be nice to also add and show what the upper bound of metric is if trained on the training dataset (target domain) when it exists.

**Summary Of The Paper:**

The paper focuses on domain-adaptation of ranking problem.
They show generalization bound for ranking problem and propose an approach to improve adaptation to minor domain shift.
Through a set of experiments on some open-source datasets they show the effectiveness of their approach.

**Summary Of The Review:**

Overall I think it is a nice idea to improve the robustness of the model trained for re-ranking.
While, they have bounds on domain adaptation generalization, I did not verify the correctness of the proof.
The experiments demonstrate superiority of their approach across three datasets.
Provided the questions mentioned in the above section is addressed, I would be inclined to accept the paper.

---

> ### Author Response · Authors · 2022-11-18
> **Response to Reviewer U7gj (Part 1 of 2)**
>
> Thank you for your helpful comments!  We have address the grammatical issues in our thorough revision, and hope that the following addresses the concerns in the review:
>
> -   **The writing in approach section blends both previous work and their approach making a bit hard to discern their contribution.**
>
>     We have revised Sections 3 and 4 (theory and method) to highlight the main theoretical and technical novelty: our domain adaptation bound is under listwise ranking metrics e.g. MRR and NDCG, and we use transformers in ListDA as the discriminator to model list-like inputs.  While in retrospect, the change may seem technically incremental, but from a conceptual perspective, this is fundamentally different from prior approaches that model inputs at the item level.
>
> -   **Why was the better negative sampling not used for other approaches...**
>
>     The better negative sampling strategy is actually used for both ItemDA and ListDA, and is not applicable on other approaches that do not involve adversarial training.  We have revamped the respective section (now Appendix B.2) on describing the construction of example lists.  In short, negative sampling from rank 300-1000 BM25-retrieved results is only used when constructing **source domain lists**, and only involved in the computation of **adversarial loss**.  The procedure requires a reliable pre-trained ranker, hence is not applicable to the target domains.  It is not involved in ranking loss for source domain supervised training because decreased performance was observed in our preliminary experiments, likely due to the exclusion of "hard" negatives.
>
> -   **It is unclear how their approach might be affected by a better retriever that could be encoder-based.**
>
>     Thank you for the remark: due to resource limitations, we did not pursue additional experiments with more sophisticated retrievers, e.g., based on dual encoders, but we expect that a better retriever would improve the performance of all approaches and the improvement brought by our method would be orthogonal to different retrievers as we do not heavily rely on information from the retriever such as the retriever scores.
>
>     The primary focus of our work is not to challenge SOTA but to lay a solid foundation for domain adaptation for ranking problems.  The existing set of experiments and results, in our opinion, is largely effective at empirically verifying our theory and discussions.  But we agree that further experiments with another retriever can deepen our understanding.  We will leave it for future work.
>
> -   **The model training is a bit confusing, should the adversarial training be done for every new target distribution or can it be trained once and re-used?**
>
>     Thank you for raising this concern: adversarial training is performed for each new target distribution (hence "domain adaptation", as opposed to "domain generalization").  This remark is added in the revision.
>
> -   **Q-D generation is a bit questionable...**
>
>     We have cleaned up the description in Section 5 (experiments) for the QGen query generation procedure.  QGen is only involved in our passage reranking experiments because of the lack of training queries from the target datasets.  The query generator is only trained on MS MARCO source domain (abundant) annotated relevant q-d pairs (following [1]), and applied to synthesize queries on the target domains in a zero-shot manner.
>
>     QGen is shown to be effective in [1] with decent generated query quality even on target domain with little to no overlapping with the source domain (Table 7).  In Section 5.2 and Fig. 2, we investigate the effects of the number of target domain documents on ListDA.  It is, however, definitely an interesting direction to explore whether a domain-adapted QGen can provide even better performance.  Since this could be non-trivial, we believe it is out of the scope of this paper and we leave it to future study.
>
> -   **There are assumptions made in section 3 for the proof, but not all assumptions are justified...**
>
>     Thank you for raising this concern: we have revised and further elaborated on all assumptions—the Plackett-Luce model, Lipschitz assumptions, and invertibility of the feature map—and commented on why they are required by our theory (namely the technical hardness).

---

> > ### Author Response · Authors · 2022-11-18
> > **Response to Reviewer U7gj (Part 2 of 2)**
> >
> >
> > -   **Show what the upper bound of metric is if trained on the training dataset (target domain) when it exists...  Not sure if the results are reproducible with their code.**
> >
> >     Thank you for the suggestion: including upper bounds (namely, target domain supervised training results) would indeed be informative for assessing the performance of domain adaptation methods, however, two out of the three passage reranking target domains considered in Section 5 (experiments) do not contain any labeled training data.
> >
> >     To this end, we have performed a new set of domain adaptation experiments on the Yahoo! Learning to Rank Challenge v2.0 dataset, a web search ranking task on numerical data that contains labeled training data on both the source and target domains.  In Table 5 of Appendix C, zero-shot, domain adaptation and supervised results are included and discussed.  The findings are consistent with those from the passage reranking experiments.  We exhaustively tuned the hyperparameters for all methods on this dataset, and plan to release the code on the Yahoo! LETOR experiments after the anonymity period for the purpose of reproducibility.
> >
> > -   **There are some minor grammatical errors...**
> >
> >     Thank you for pointing this out!  We have thoroughly revised our draft, including the paragraph in question.
> >
> > [1]  Zero-shot Neural Passage Retrieval via Domain-targeted Synthetic Question Generation. 2021.

---

> > > ### Comment · Reviewer_U7gj · 2022-12-02
> > > **Thanks for the response**
> > >
> > > Hello thank you for the detailed response and addressing the comments. I will re-evaluate and update based on comments .

---

### Official Review · Reviewer_3EyT · 2022-10-25

**Confidence:** 4
**Correctness:** 3
**Technical Novelty And Significance:** 3
**Empirical Novelty And Significance:** 2
**Recommendation:** 5

**Clarity, Quality, Novelty And Reproducibility:**

This is generally a good paper and authors provide all the required and relevant material for the both the theoretical and experimental part. For the latter though, I cannot really comment on the reproducibility.

As explained previously, this is an extension of an existing theoretical framework. There is some originality though for the specific task.

**Strength And Weaknesses:**

- This is an interesting problem/setting as in many cases it is very expensive to collect labeled data.
- The authors extend work on domain adaptation on the l2r setting which is an important task in many applications
- The results seem to be promising although only applied in the textual domain.

- The proposed approach is not so tightly connected with the theoretical part. Indeed the authors use standard adversarial setting for domain adaptation. I would propose the authors to better describe the connection. These approaches are already presented, see for example Martin Arjovsky, Soumith Chintala, and Léon Bottou. Wasserstein generative adversarial networks. ICML 2017.
Ganin et al., Domain-adversarial training of neural networks. Journal of Machine Learning Research, 2016.

- I think the authors should discuss this work "On Learning Invariant Representations for Domain Adaptation" regarding the optimal joint risk.

- Generally, while the experimental part show promising results, the fact that is only applied in the textual domain makes it weak. Also, already the zero-shot baseline has a good performance, and to be honest I am not convinced how much this small difference would make in a production system. I am just curious here if we carefully tune the zero-shot baseline what the result would be.
- Is there any reason to restrict to softmax listwise loss function? Did you try other ones? For example BoltzRank?

**Summary Of The Paper:**

The authors propose a domain-invariant adaptation approach for listwise learning to rank (l2r). The authors extend a bound from the literature to l2r setting. Then a typical approach for domain adaptation is used with adversarial learning. The authors experiment in the textual domain with one source and three target domains. They compare with two baselines and two relevant approaches showing that the proposed method outperforms them.

**Summary Of The Review:**

Generally a good paper, with a lack of connection between the theoretical and the method part. It builds upon an existing framework and extends for the l2r framework. The experimental part is weak in my opinion as it only applied on the textual domain and is hard to asses if this approach would work in other domains. Already the baseline works quite well.

---

> ### Author Response · Authors · 2022-11-18
> **Response to Reviewer 3EyT (Part 1 of 2)**
>
> Thank you for your helpful comments!  And we hope that the following addresses the concerns in the review:
>
> -   **The authors extend work on domain adaptation on the l2r setting...  These approaches are already presented [7, 8, 9]...**
>
>     The key distinction of our proposed method—learning listwise domain-invariant representations, or ListDA—is that the invariant representations it learns are at the list level, but not at the item level.  Indeed, it uses adversarial training as the optimization technique, but differs from prior work in GAN [7, 8], domain adaptation for classification [9, 13, 14], and domain adaptation for ranking [10, 11, 12] where the distributions being models and invariant representations learned are at the item level.  Specifically, we model list-like data with transformers, which from a conceptual perspective is fundamentally different from prior approaches that model inputs at the item level.  As pointed out in Section 4 (method), the approach taken in prior work is inappropriate for list data in ranking problems.
>
>     In particular, achieving representation invariance at the item level—performed in [10, 11, 12] and which are similar in terms of model architecture and objective design to [7, 8, 9, 13, 14]—does not imply invariance at the list level.  Instead, our theory suggests that list level invariance is more appropriate under listwise metrics, which is also empirically verified by our experiments.
>
> -   **[Theorem 2] is an extension of an existing theoretical framework.  The proposed approach is not so tightly connected with the theoretical part...**
>
>     Our theory (Theorem 2) is indeed based on the domain adaptation framework established in [1], and takes a seemingly similar form as that of [2] (Theorem 1).  However, we included Theorem 1 in our presentation (domain adaptation bound for binary classification) only to facilitate the understanding and interpretation of our result.  The technique with which we established Theorem 2 diverges significantly from those of [2, 1], which we now highlight and elucidate in Section 3 (theory) of our revised version.  In addition, existing results on domain adaptation for classification cannot be extended to give a statement for listwise ranking metrics (e.g., MRR and NDCG), due to the looseness.
>
>     Specifically, the technical hardness of our result comes from the list structure of the data and the metrics in ranking problems, as well as the well-known discontinuity of ranking metrics (including RR and NDCG) w.r.t. model outputs.  These issues are addressed using techniques that have not appeared in prior work on domain adaptation.
>
>     The main insight from our result is that domain adaptation generalization under listwise ranking metrics is achieved if the feature representations are domain-invariant at the list level, not at the item level.  Since representation invariance at the item level does not imply invariance at the list level, it means that listwise invariant representation learning is more appropriate for domain adaptation for ranking under listwise metrics.
>
>     Our method, ListDA, reflects the above insight, and is an instantiation of the domain adaptation approach suggested by Theorem 2, that domain transferability could be improved via learning a domain-invariant feature representation (reduce W1 between source and target marginal feature distributions) and training the ranker using source domain data on top of the shared and invariant feature representations.
>
>     Thank you for the comment: we have revised Section 3 (theory) to emphasize the connection between our theory and our method, and improved the flow.
>
> -   **The results seem to be promising although only applied in the textual domain...  the fact that is only applied in the textual domain makes it weak...  cannot really comment on the reproducibility.**
>
>     We have performed a new set of domain adaptation experiments on the Yahoo! Learning to Rank Challenge v2.0 dataset, a web search ranking task on numerical data.  The experiment details are provided in Appendix C of the revised draft, and the results are consistent with those from the passage reranking experiments and also support our earlier findings.  We exhaustively tuned the hyperparameters for all methods on this dataset, and plan to release the code on the Yahoo! LETOR experiments after the anonymity period for the purpose of reproducibility.

---

> > ### Author Response · Authors · 2022-11-18
> > **Response to Reviewer 3EyT (Part 2 of 2)**
> >
> >
> > -   **"On Learning Invariant Representations for Domain Adaptation" regarding the optimal joint risk.**
> >
> >     Thank you for pointing out this work [3], which establishes that for classification problems, Theorem 1 (domain adaptation bound for binary classification) admits a lower bound under perfect feature alignment and source accuracy when there are distribution shifts in class priors between source and target domains.  This happens in scenarios where the optimal joint risk (which is unknown in unsupervised domain adaptation) is nonzero under the feature transformation.  Although our method, as well as all prior unsupervised domain adaptation approaches based on invariant representation learning, cannot guarantee that the optimal joint risk remains low under the learned feature map, it does not preclude their empirical success in many fields, including in our experiments.
> >
> >     Theory-wise, the findings of [3] only apply to classification problems and not ranking.  There may be a similar lower bound for Theorem 2 when the marginal distributions of ground-truth relevance scores differ between source and target domains.  A discussion on this is now included in the related works section in our revision.  As our work is focused on establishing an initial theoretical foundation for domain adaptation for ranking problems, and in particular, marking the distinction between representation learning at the list level and the item level, this direction was not pursued.  But it is our hope that our results could inspire more work in this arguably underexplored area.
> >
> > -   **If we carefully tune the zero-shot baseline what the result would be.**
> >
> >     Thank you for the suggestion.  The hyperparameters used in our zero-shot baselines, in particular, the learning rate of the reranker, are already carefully tuned from a grid search for each target domain respectively.  We have also added the zero-shot performance from the parameter sweep in Fig. 4 on page 20, and include the zero-shot results of ranking models with a similar size in prior work below:
> >
> >     | NDCG@10 (zero-shot)                             | Robust04 | TREC-COVID | BioASQ |
> >     | ----------------------------------------------- | -------- | ---------- | ------ |
> >     | Ours                                            | 0.5340   | 0.8200     | 0.5542 |
> >     | [4], same model but with *better* MS MARCO data | -        | 0.7896     | 0.5627 |
> >     | [5]                                             | 0.5016   | 0.7775     | 0.5249 |
> >     | [6]                                             | 0.475    | 0.757      | 0.523  |
> >
> >     In addition, in our new set of experiments on Yahoo! LETOR, the hyperparameters of all methods, including zero-shot, are tuned individually and exhaustively from a wider range of candidate settings.  Hence the results of Table 5 also be referred to for comparisons between zero-shot and domain adaptation methods.
> >
> > -   **Is there any reason to restrict to softmax listwise loss function?**
> >
> >     Thank you for the suggestion: we primarily use the listwise softmax cross-entropy loss in our experiments because of their good empirical performance compared to alternative ranking losses [4].  We have added a new set of experiments in Table 2(a) on page 19 using the pairwise logistic ranking loss on the Robust04 dataset; the findings are consistent with earlier results, but overall the performance is less competitive than softmax cross-entropy.  The empirical findings are expected to hold with other ranking losses.
> >
> > [1]  Analysis of Representations for Domain Adaptation. 2007.
> > [2]  Wasserstein Distance Guided Representation Learning for Domain Adaptation. 2018.
> > [3]  On Learning Invariant Representations for Domain Adaptation. 2019.
> > [4]  RankT5: Fine-Tuning T5 for Text Ranking with Ranking Losses. 2022.
> > [5]  No parameters left behind: Sensitivity guided adaptive learning rate for training large transformer models. 2022.
> > [6]  BEIR: A Heterogeneous Benchmark for Zero-shot Evaluation of Information Retrieval Models. 2021.
> > [7]  Generative Adversarial Nets. 2014.
> > [8]  Wasserstein Generative Adversarial Networks. 2017.
> > [9]  Domain-Adversarial Training of Neural Networks. 2016.
> > [10]  Cross Domain Regularization for Neural Ranking Models using Adversarial Learning. 2018.
> > [11]  Domain Adaptation for Enterprise Email Search. 2019.
> > [12]  Zero-Shot Dense Retrieval with Momentum Adversarial Domain Invariant Representations. 2022.
> > [13]  Learning Transferable Features with Deep Adaptation Networks. 2015.
> > [14]  Conditional adversarial domain adaptation. 2018.

---

### Official Review · Reviewer_SYBU · 2022-10-27

**Confidence:** 4
**Clarity, Quality, Novelty And Reproducibility:** Please refer to the Strength and Weak…
**Correctness:** 4
**Technical Novelty And Significance:** 3
**Empirical Novelty And Significance:** 2
**Recommendation:** 6

**Strength And Weaknesses:**

Strengths.
- The paper is well written and technically sound
- The paper proposes a domain adaptation method for learning to rank tailored for the listwise scenario. Compared to existing methods, the proposed approach seeks invariance at the list representation level instead of the item level.
- Theoretically, the authors derive a domain adaptation generalization bound for the listwise learning to rank case.

Weaknesses.

- The experiments are weak. (I) Evaluations are curried out on one task and one data type (text) only. (II) The list of baselines can be improved, for instance by including other domain adaptation learning to rank approaches, such as the one mentioned in the related work section. (III) Some results are not reported for all datasets and baselines (e.g., the results of figures 2 and 3).
- The technical novelty of the proposed objective is somewhat limited.

Additional comments.
I would recommend presenting section 4 first followed by the current section 3 as supportive theoretical results for the proposed objective.


**Summary Of The Paper:**

This paper focuses on Domain Adaptation (DA) in the context of learning to rank. The authors follow the invariant representation learning approach, which they extend to the listwise learning to rank scenario. The final objective consists of a listwise ranking loss and an adversarial loss encouraging invariance across domains.  The authors also derive a domain adaptation generalization bound for listwise learning to rank scenario. The proposed method is evaluated on the passage/text re-ranking task.

**Summary Of The Review:**

Please refer to the Strength and Weaknesses section above.

---

> ### Author Response · Authors · 2022-11-18
> **Response to Reviewer SYBU**
>
> Thank you for your helpful comments!  And we hope that the following addresses the concerns in the review:
>
> -   **(I) Evaluations are carried out on one task and one data type (text) only.**
>
>     We have performed a new set of domain adaptation experiments on the Yahoo! Learning to Rank Challenge v2.0 dataset, a web search ranking task on numerical data.  The experiment details are provided in Appendix C of the revised draft, and the results are consistent with those from the passage reranking experiments and also support our earlier findings.  We exhaustively tuned the hyperparameters for all methods on this dataset, and plan to release the code on the Yahoo! LETOR experiments after the anonymity period for the purpose of reproducibility.
>
> -   **(II) The list of baselines can be improved... by including other domain adaptation learning to rank approaches... mentioned in the related work section.**
>
>     Our method is primarily targeted for unsupervised domain adaptation, and from our study of prior works, we found that the variety of approaches for learning to rank is limited.  As summarized in the related work section, for general ranking problems, the main existing adaptation method is invariant representation learning on the item level [1, 2, 3, 4], to the best of our knowledge.  For the special case of text ranking, recent work leverages query generation to perform pseudolabel training for domain adaptation [5, 6, 7].  These prior approaches are already represented by the baseline methods of **ItemDA** and **QGen PL** in our experiments, respectively.
>
>     The primary focus of our work is not to challenge SOTA but to lay a solid foundation for domain adaptation for ranking problems; most importantly, we show theoretically and empirically that learning representations at the list level (ListDA) is more appropriate than at the item level (ItemDA).  Indeed, it is our hope that our results could inspire more work in this arguably underexplored area.
>
> -   **(III) Some results are not reported for all datasets and baselines.**
>
>     We have added the results that were missing from BioASQ in revised Figs. 2 and 3, and updated the discussions.
>
> -   **The technical novelty of the proposed objective is somewhat limited.**
>
>     We have revised Section 4 (proposed method) to highlight the main novelty of ListDA—the use of transformers as the discriminator to model list-like inputs.  While in retrospect, the change may seem technically incremental, but from a conceptual perspective, this is fundamentally different from prior approaches that model inputs at the item level.  The proposed objective in its current form also allows us to perform controlled comparison experiments to prior work (ItemDA), and is appropriate for empirical validations of our theory and discussions.
>
> -   **Presenting section 4 first followed by the current section 3...**
>
>     Thank you for the comment: presenting the method first followed by theoretical results as support is a good alternative, although the current organization of presenting the theory followed by method, in our opinion, better reflects our main contributions.  We have revised Section 3 (theory) to emphasize the connection between our theory and our method, and improved the flow—we view our method as an instantiation of the approach suggested by our theory, that we hope may inspire future empirical investigations.
>
> [1]  Cross Domain Regularization for Neural Ranking Models using Adversarial Learning. 2018.
> [2]  Domain Adaptation for Enterprise Email Search. 2019.
> [3]  Zero-Shot Dense Retrieval with Momentum Adversarial Domain Invariant Representations. 2022.
> [4]  Domain-Adversarial Training of Neural Networks. 2016.
> [5]  Zero-shot Neural Passage Retrieval via Domain-targeted Synthetic Question Generation. 2021.
> [6]  GPL: Generative Pseudo Labeling for Unsupervised Domain Adaptation of Dense Retrieval. 2022.
> [7]  Few-Shot Text Ranking with Meta Adapted Synthetic Weak Supervision. 2021.

---

### Author Response · Authors · 2022-11-18
**Revision Summary**

We thank the reviewers for the time and the helpful comments!  We have thoroughly revised our draft to address the concerns in the review, and improved clarity and presentation.

The main changes are:

-   Added a new set of unsupervised domain adaptation experiments on the Yahoo! Learning to Rank Challenge v2.0 dataset, a web search ranking task on numerical data (Appendix C, page 22).
-   Added passage reranking experiments with the pairwise logistic ranking loss on the Robust04 dataset (in addition to existing ones with listwise softmax cross-entropy; Table 2(a), page 19).
-   Added case studies from the passage reranking experiments to provide a qualitative understanding of the advantages of our method (Tables 8 and 9, pages 25–26, and discussions in Appendix B, page 19).
-   Revised Sections 3 and 4 (theory and method) to highlight the technical novelty of our work and the distinctions from existing results.
-   Cleaned up and revamped the sections on further details of the implementation and setup of the passage reranking experiments (mainly Appendix B.2, pages 20–22).

---

### Decision · Program_Chairs · 2023-01-20

**Decision:**

Reject

**Justification For Why Not Higher Score:**

Details have been described in the meta-review.

**Justification For Why Not Lower Score:**

N/A

**Metareview: Summary, Strengths And Weaknesses:**

In this paper, the authors proposed a new transfer learning method for ranking problems.

As this is a borderline paper, an AC-reviewer meeting was conducted. First, reviewers are satisfied with additional experiments. Therefore, experimental results are no longer a concern about this submission. However, there are some major concerns that remain: 1) the application of the proposed method is somewhat narrow, which may not be applied to general ranking problems. 2) the connection between the theoretical study and the proposed methodology is not clear. They look independent. 3) the theoretical analysis is a straight-word extension of existing analysis to the specific ranking problems, which lacks new techniques. 4) More discussions are needed for a comparison between listwise and pairwise ranking methods.

While promising experimental results are reported, the aforementioned concerns need to be addressed. The authors are encouraged to revise the paper based on the reviewers' comments for future submission.

**Summary Of Ac-Reviewer Meeting:**

The summary of AC-reviewer meeting has been described in the detailed meta-review.